# Exercise-induced histone lactylation in monocyte-derived macrophages restores cardiac immune homeostasis and function in sepsis-induced cardiomyopathy

Shuo Sun[1,6,7,8], Chaojie Lai[1,8], Chengchen Huang[1], Xinrong Ren[2], Tingyu Zhang[1], Juan Zou[3], Yiping Tong[2], Qingyan Zhou[1], Jiangting Lu[1], Zhida Shen[1], Wentao Chen[1], Ruilin Wang[1], Nikola Rabrenovic[1], Xingwu Wang [3], Boxuan Ma [1] ✉, Junbin Qian [4,5] ✉, Guosheng Fu[1] ✉ & Min Shang [1,2] ✉

An active lifestyle protects against cardiovascular diseases, yet its mechanisms in modulating the cardiac immune environment and preserving cardiac function remain unclear. Here, we identify a subpopulation of monocyte-derived cardiac macrophages, termed iNOS+Arg1+ macrophages, which simultaneously express pro-inflammatory and pro-reparative genes in exercised male mice. Inhibiting either pro-inflammatory iNOS or pro-reparative Arg1 in these macrophages counters the exercise-induced cardiac function preservation. Mechanistically, exercise enhances glycolysis in monocytes, increasing lactate production and driving histone lactylation at H3K18, mediated by p300 as the relevant lactyltransferase and counterbalanced by HDAC2 as deacetylase. H3K18la accelerates the transition of cardiac macrophages to a pro-reparative state, restoring immune homeostasis and preserving cardiac function. Notably, human monocytes from physically active individuals exhibit elevated levels of Pan-Kla and H3K18la compared to those from sedentary individuals. Importantly, adoptive transfer of highly histone-lactylated monocytes restores cardiac function in sepsis-induced cardiomyopathy, which might translate into a promising therapeutic strategy for cardiomyopathy.

Sepsis is a leading cause of death in the intensive care unit, characterized by multiple organ dysfunctions triggered by host infection[1,2]. Lipopolysaccharide (LPS), a potent endotoxin derived from Gram-negative bacteria, plays a pivotal role in the pathogenesis of sepsis and its associated organ failure[3–5]. Among septic patients, up to 60% experience cardiac dysfunction[6], referred to as sepsis-induced cardiomyopathy (SICM). SICM is clinically manifested by reduced ejection fraction, left ventricular dilation, and early reversible myocardial

[1]Zhejiang Key Laboratory of Cardiovascular Intervention and Precision Medicine, Engineering Research Center for Cardiovascular Innovative Devices of Zhejiang Province, Department of Cardiology, Sir Run Run Shaw Hospital, School of Medicine, Zhejiang University, Hangzhou, China. [2]School of Basic Medical Sciences and Forensic Medicine, Hangzhou Medical College, Hangzhou, Zhejiang, China. [3]Laboratory of Cell Fate and Metabolic regulation, Shenzhen campus of Sun Yat-sen University, Shenzhen, China. [4]Zhejiang Key Laboratory of Precision Diagnosis and Therapy for Major Gynecological Diseases, Women's Hospital, Zhejiang University School of Medicine, Hangzhou, China. [5]Zhejiang Provincial Clinical Research Center for Child Health, Hangzhou, China. [6]Present address: Jining Medical University, Jining, China. [7]Present address: School of Life Sciences, Jining Medical University, Rizhao, China. [8]These authors contributed equally: Shuo Sun, Chaojie Lai. ✉e-mail: boxuanma@zju.edu.cn; dr_qian@zju.edu.cn; fugs@zju.edu.cn; minshang@zju.edu.cn

dysfunction[7–9]. Notably, SICM is a major contributor to multiple organ dysfunction and significantly impacts the prognosis and mortality of sepsis patients, with mortality rates approximately three times higher compared to sepsis patients without cardiac dysfunction[10,11].

Cardiac immune homeostasis plays a critical role in reversing myocardial dysfunction and facilitating cardiac rehabilitation after septic stress[5,12], with macrophages being the predominant immune cells recruited in the septic heart[5]. Macrophages are highly plastic and heterogeneous populations, with distinct subsets exhibiting divergent functions modulated by different immune environments[13–16]. We and others have previously shown that metabolic reprogramming orchestrates recruited macrophages to perform essential functions such as cell debris clearance and tissue remodeling in conditions like cancer, muscle degeneration, and sepsis[15–18]. Thus, understanding the macrophage subsets and the signals that induce them may provide strategies to restore cardiac homeostasis in the septic heart.

Regular physical exercise mitigates the metabolic and immunological adaptations that contribute to various inflammatory-related diseases[19,20], thereby protecting against cardiovascular diseases (CVD) and cancer[21–23]. Regular exercise mobilizes and redistributes natural killer (NK) cells to suppress tumor growth[24]. Furthermore, exercise diminishes hematopoietic progenitor cell proliferation and leukocyte production, which reduces inflammation in atherosclerosis and improves its outcomes[25]. The beneficial effects of regular physical exercise on the immune system[26–28] and CVD progression[22,29,30] are well studied. However, the effects of physical exercise on the cardiac immune landscape remain to be fully elucidated. Therefore, we investigated whether and how voluntary running pre-trains cardiac macrophages to preadapt and protect cardiac functions in sepsis.

In this study, we find that voluntary running preserves cardiac function and immune homeostasis in sepsis. To investigate how exercise modulates cardiac immune cells in SICM, we generate a cardiac immune single-cell atlas, which serves as a valuable resource for identifying therapeutic strategies that mimic exercise-induced benefits in SICM. The immunoregulatory benefit is attributed to histone lysine lactylation (Kla), specifically at H3K18la, which is mediated by p300 and HDAC2 as the lactyltransferase and delactylase, respectively. Exercise-induced lactate triggers H3K18la, effectively pre-training monocyte-derived macrophages to preadapt to sepsis before their cardiac recruitment in SICM. Finally, through the adoptive transfer of lactate-educated monocytes, we demonstrate that this approach preserves cardiac function in SICM, offering a potential cellular therapy to harness the cardioprotective effects of exercise.

## Results

### Regular physical exercise safeguards cardiac function against SICM by modulating the cardiac immune microenvironment

To evaluate the impacts of regular physical exercise on septic cardiac function, mice were provided with running wheels for voluntary exercise for three months. Subsequently, mice were injected intraperitoneally with either lipopolysaccharide (LPS) or an equivalent volume of phosphate-buffered saline (PBS) to mimic endotoxin-septic shock and induce sepsis-induced cardiomyopathy (SICM) in sedentary (Sed) and exercised (Exe) groups (Fig. 1A, B). Our observations revealed that regular exercise protected against LPS-induced sepsis, as shown by a reduced decline in body temperature and body weight 18 h post-injection (Fig. 1C, D). While LPS injection impaired cardiac function in sedentary mice, exercised mice maintained preserved cardiac function, with improvements in ejection fraction (EF) and fractional shortening (FS) by approximately 21% and 34%, respectively (Fig. 1E–G). Consistently, TUNEL and dihydroethidium (DHE) staining showed reduced apoptosis and oxidative damage in the hearts of mice with SICM (Fig. 1H–K). In contrast, body temperature, body weight, and cardiac function were comparable between Sed and Exe mice in

the PBS control group (Fig. 1C–K). Collectively, these results suggest that voluntary running preserves cardiac function during sepsis.

To uncover the protective mechanisms of regular exercise on cardiac function during sepsis, we conducted bulk RNA-sequencing analysis on the left ventricles of PBS (Sed), PBS (Exe), LPS (Sed), and LPS (Exe) mice. Principal components analysis (PCA) revealed that exercise-induced transcriptional changes were more pronounced in the SICM groups compared to the PBS controls (Fig. 1L). Differentially expressed gene (DEG) analysis identified 447 DEGs between LPS (Exe) and LPS (Sed) mice, compared to only 14 DEGs between PBS (Exe) and PBS (Sed) mice (Fig. 1M). Among the 14 DEGs, 2 genes (*Marco* and *Gbp11*) were upregulated by exercise, while 12 genes (*Cd300lf*, *Fgr*, *Mmp8*, *Ccl22*, *Vwa3b*, *Cd177*, *Hspa1a*, *Hsd11b2*, *Slc7a11*, *Fndc7*, *Nr4a3*, *Mcemp1*) were downregulated (Fig. 1M). In contrast, among the 447 DEGs affected by exercise in SICM, exercise suppressed 111 genes upregulated by SICM, including *Cxcl3*, *Il6*, *Ccl2*, and *Cd14*, which are primarily involved in pathways such as response to chemokine, chemotaxis, and cell chemotaxis. On the other hand, exercise activated 274 genes downregulated by SICM, including those associated with action potential, cardiac muscle contraction, lipid biosynthetic process, and so on (Fig. 1N). These findings suggest that regular exercise mitigates the excessive immune activation caused by SICM while restoring transcriptional pathways essential for cardiac function that are disrupted by SICM. To further investigate this, we measured the serum concentrations of IL-1β and TNF-α and found that physical exercise effectively attenuated LPS-induced pro-inflammatory responses (Fig. 1O, P). While the total CD68+ macrophage infiltration remained unchanged, the Arg1+ pro-reparative macrophages in exercised septic hearts increased by 104% compared to sedentary septic hearts (Fig. 1Q–S). These results suggest that the cardioprotective effects of regular exercise during sepsis are likely attributed to the modulation of the cardiac immune microenvironment.

### Voluntary running orchestrates the septic cardiac immune microenvironment

To further characterize the cardiac immune microenvironment in exercised septic hearts, and given the limited differences observed between PBS (Sed) and PBS (Exe) groups, we isolated the CD45+ cell populations from PBS (Sed), LPS (Sed) and LPS (Exe) mice and conducted single-cell RNA sequencing (scRNA-seq) analysis using the 10X Genomics platform (Fig. 2A). A total of 50,471 quality-control-positive cells were utilized for further analysis, generating 40,892 CD45+ cells and 19,184 expressed genes (Table S1). Unbiased clustering analysis unveiled 13 distinct clusters of cardiac immune populations, including macrophages (*Adgre1* and *C1qa*), monocytes (*Plac8* and *Ly6c2*), conventional dendritic (cDC) cells (*Cd209a* and *Fscn1*), plasmacytoid dendritic (pDC) cells (*Siglech*), neutrophils (*Mmp9* and *Csf3r*), natural killer (NK) cells (*Nkg7* and *Klrb1c*), CD4+ and CD8+ T cells (*Cd3g*, *Cd4* and *Cd8a*), gamma delta T (gdT) cells (*Trdc* and *Trdv4*), B cells (*Cd79a* and *Cd19*), innate lymphoid cells (ILC) type 2 (*Gata3* and *Rora*), mast cells (*Gata2* and *Mcpt8*) and cycling cells (*Mki67* and *Stmn1*) (Fig. 2B, C). Consistent with previous findings[12], sepsis induced the infiltration of monocytes and neutrophils into the heart (Fig. 2D, E). Importantly, regular physical exercise significantly enhanced the infiltration of monocytes into septic hearts (Fig. 2F). Immune infiltration analysis based on bulk RNA sequencing data aligned with these observations, furthermore, showed no significant differences between the PBS (Sed) and PBS (Exe) groups (Fig. S1). Taken together, these data hint that regular physical exercise modulates septic cardiac monocyte-derived macrophages to orchestrate the cardiac immune microenvironment.

### Voluntary running modulates cardiac monocyte and macrophage subsets in SICM

Given that monocyte and macrophage populations constitute nearly half of all immune cells in the septic heart, we focused our

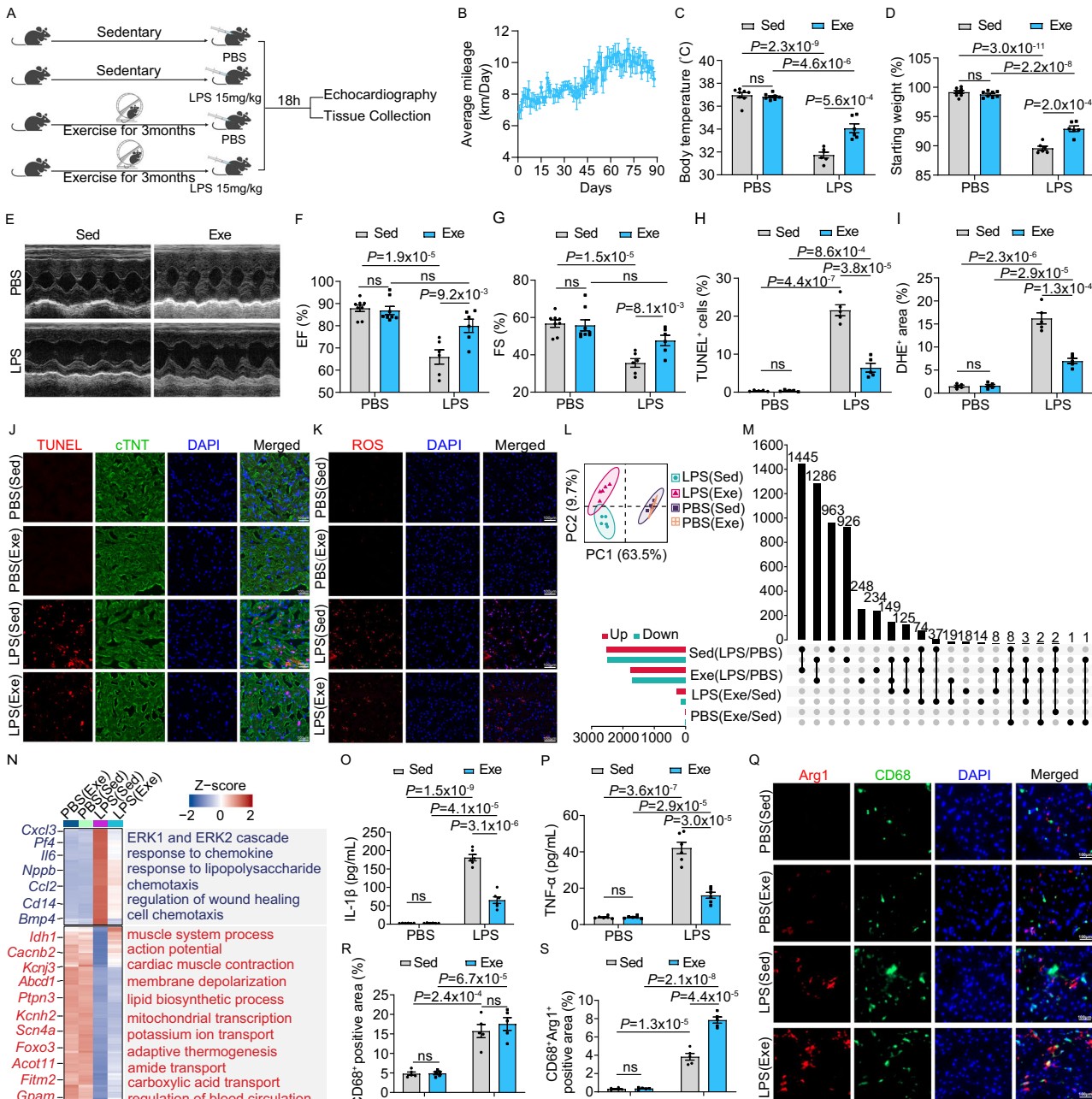

**Fig. 1 | Regular physical exercise safeguards cardiac function against SICM by modulating the cardiac immune microenvironment.** Schematic representation of the mice treatment (**A**) and average voluntary running distance (km/day) (**B**). The analysis includes a total of 14 mice. Figure created in BioRender, Shang, M. (2025) tc3zfga. Body temperature (**C**) and percent change in body weight (**D**) of 18 h post intraperitoneal injection of PBS or LPS in sedentary (Sed; n = 8 for PBS, n = 6 for LPS) and exercise (Exe; n = 8 for PBS, n = 6 for LPS) mice. Representative echocardiography images (**E**) and quantification of EF% (**F**) and FS% (**G**) on 18 h post PBS or LPS i.p injection on Sed (n = 8 for PBS, n = 6 for LPS) and Exe (n = 8 for PBS, n = 6 for LPS) mice. Representative images and quantification of TUNEL (**H, J**) and DHE (**I, K**) in heart sections 18 h post intraperitoneal injection of PBS or LPS in Sed (n = 5 for PBS, n = 5 for LPS) and Exe (n = 5 for PBS, n = 5 for LPS) mice. The experiment shows representative values from 3 independent experiments. **L** Principal component analysis (PCA) of the transcriptomes of PBS (Sed), PBS (Exe), LPS (Sed), and LPS

(Exe) groups. **M** Upset plot displays the number and overlap of DEGs between LPS (Sed) and PBS (Sed), LPS (Exe) and PBS (Exe), LPS (Exe) and LPS (Sed), PBS (Exe) and PBS (Sed) groups. **N** Heatmap of differential expression patterns. Samples from different groups are illustrated above, and representative genes with associated GO terms are illustrated. ELISA analysis of IL-1β (**O**) and TNF-α (**P**) in Sed and Exe mice. The analysis was performed on 6 samples per group. Quantitative of infiltrated macrophages (CD68⁺) (**R**) and CD68⁺Arg1⁺cells (**S**), and representative images of immunofluorescent staining of CD68 (green) and Arg1 (red) in hearts of mice (**Q**) with indicated treatments. The analysis was performed on 5 samples per group from 3 independent experiments. Data are represented as mean ± SEM. Statistical analysis was performed using two-tailed unpaired t-tests (**C, D, F–I, O, P, R, S**). NS, not significant. Scale bar = 100 μm (**J, K, Q**). Source data for **B–D, F–I, O, P, R**, and **S** are provided in the Source Data file.

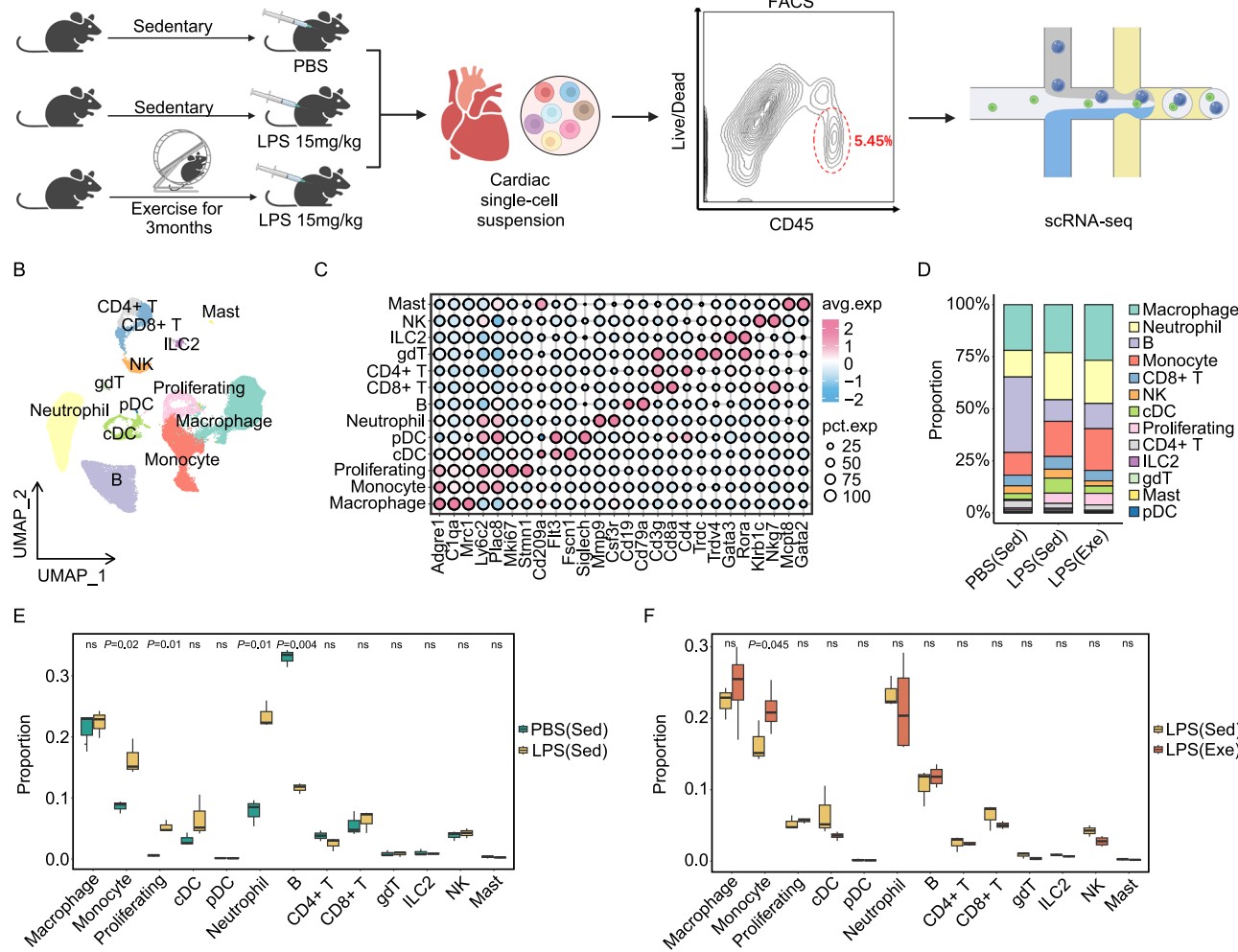

**Fig. 2 | Voluntary running orchestrates the septic cardiac immune micro-environment. A** Schematic representation of the 10X Chromium scRNA-seq workflow used to analyze CD45⁺ immune cells isolated from the left ventricles of C57BL/6 mouse hearts. Mice were either sedentary (Sed) or exercise (Exe) and received intraperitoneal injections PBS or LPS. Figure created in BioRender, Shang, M. (2025) ly8ygbn. **B** Uniform Manifold Approximation and Projection (UMAP) plot depicting 40,892 cardiac immune cells, colored according to cell type. **C** Dot plot displaying the top expressed genes in each cell population, compared to the other cell populations identified. **D** Proportions of different cell populations identified in each group. Box plot comparing the differences in the proportions of immune cell populations between the PBS (Sed) and LPS (Sed) (**E**), and LPS (Sed) and LPS (Exe) groups (**F**). Number of samples included in the analysis: PBS (Sed), n = 3; LPS (Sed), n = 3; LPS (Exe), n = 4. Data are represented as mean ± SEM. The lower whisker, lower hinge, box center, upper hinge, and upper whisker represent the minimum, lower quartile, median, upper quartile. Unpaired two-tailed t-test was used to determine the statistical significance. ns not significant. Source data for **D**, **E**, and **F** are provided in the Source Data file.

investigation on these populations to understand how voluntary running influences their recruitment and function during SICM. We began by depleting monocyte and macrophage compartments using clodronate liposomes[31–33], thereby assessing their role in the cardiac protection conferred by regular physical exercise. Consistent with our previous observations, exercise mitigated body weight loss, preserved body temperature and cardiac functions during sepsis. Notably, these beneficial effects were abolished when monocytes and macrophages were depleted, with no significant differences observed between the LPS (Sed) and LPS (Exe) mice post-depletion (Fig. 3A–E). Furthermore, monocyte and macrophage depletion abolished the reduction in apoptosis and oxidative damage in the septic heart that was previously observed in mice engaged in voluntary running (Fig. 3F–I). Altogether, our data indicate that monocytes and macrophages are essential for the cardiac protection conferred by regular exercise in SICM.

To further elucidate the contribution of monocytes and macrophages to cardiac immune homeostasis in SICM, we performed a secondary clustering analysis, identifying six subpopulations based on

subset signature genes and classic markers[34–37]. Among these, two distinct monocyte clusters and three distinct macrophage clusters were identified: Ly6c2^low Mo, Ly6c2^high Mo, Lyve1⁺ Mø (Lyve1⁺MHC II^lowCcr2⁻), MHC II^high Mø (Lyve1⁻MHC II^highCcr2⁻) and Ccr2⁺ Mø (Lyve1⁻MHC II^highCcr2⁺) (Fig. 3J). Based on Ly6c2 expression levels, monocytes were categorized into pro-inflammatory monocytes (Ly6c2^high Mo) and anti-inflammatory monocytes (Ly6c2^low Mo). Lyve1⁺ Mø primarily relied on self-renewal, MHC II^high Mø was partially replenished by monocytes, whereas Ccr2⁺ Mø was completely replenished by monocytes (Fig. 3J–L). Importantly, we identified a subpopulation adjacent to the Ly6c2^high Mo, characterized by iNOS expression (iNOS⁺ Mo). Interestingly, this subpopulation expressed both pro-inflammatory and pro-reparative genes such as *Slpi*, *Arg1*, and *Fabp5* in septic hearts (Fig. 3K). Importantly, this monocyte-derived iNOS⁺ subset was significantly elevated in septic hearts and further induced by physical exercise (Fig. 3M, Fig. S2). When comparing the relative proportions of these subclusters across different groups, we observed a significant elevation in Lyve1⁺ Mø and a significant decline

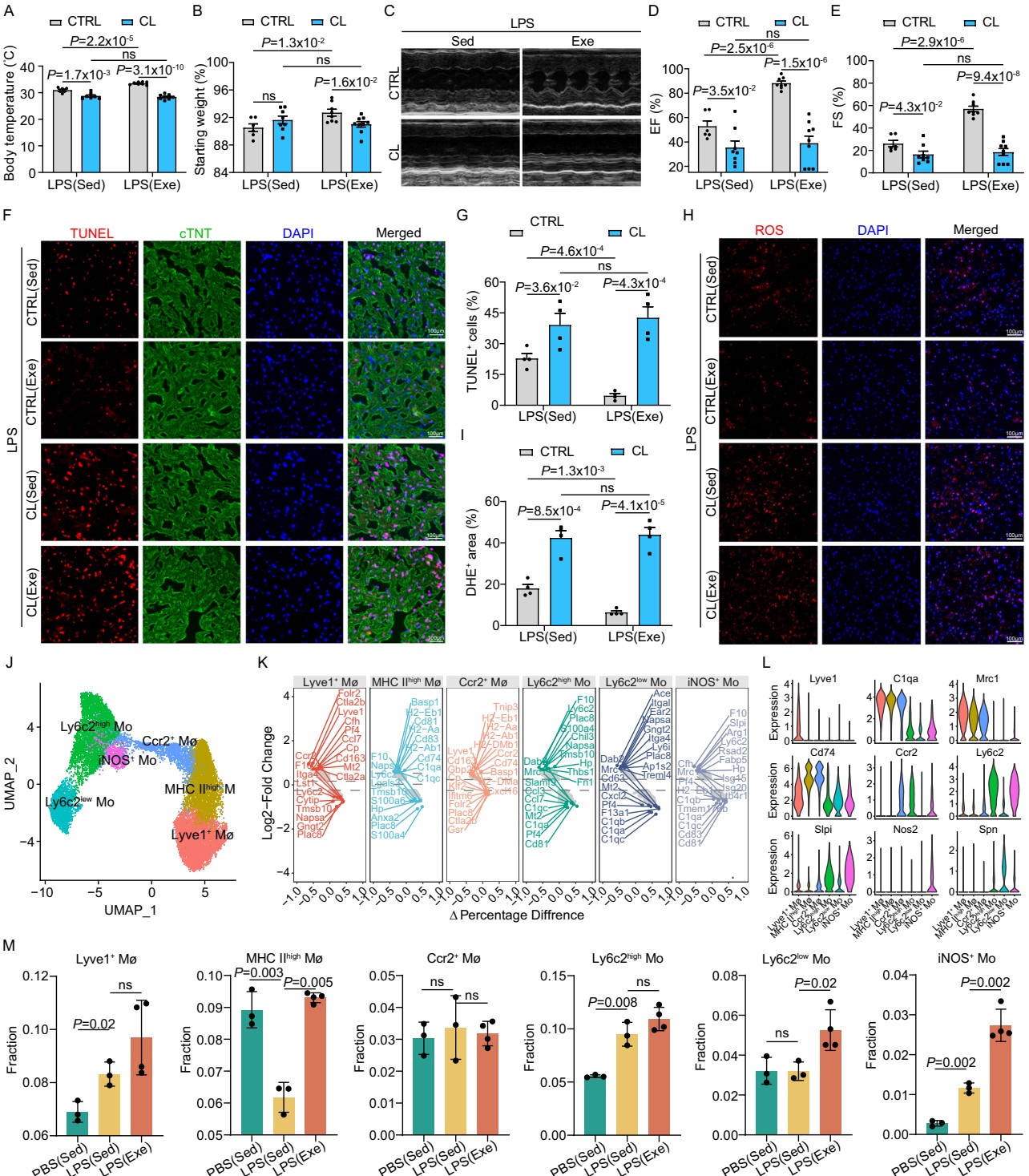

**Fig. 3 | Voluntary running modulates cardiac monocyte and macrophage subsets in SICM.** Body temperature (**A**) and percentage change in body weight (**B**) measured 18 h post LPS i.p injection in sedentary (Sed) and exercise (Exe) mice with macrophage depletion using clodronate liposomes or control liposomes (CTRL). Number of samples included in the analysis: Sed (CTRL, n = 6; CL, n = 8), Exe (CTRL, n = 8; CL, n = 9). **C–E** Representative echocardiography images (**C**) and quantification of EF% (**D**) and FS% (**E**) 18 h post-LPS i.p injection in Sed and Exe mice with clodronate liposomes or CTRL. Number of samples included in the analysis: Sed (CTRL, n = 6; CL, n = 8), Exe (CTRL n = 8; CL, n = 9). Representative images and quantification of TUNEL (**F, G**) and DHE (**H, I**) in heart sections 18 h post-LPS i.p. injection in Sed and Exe mice with clodronate liposomes or CTRL. The analysis was performed on 4 samples per group from 3 independent experiments. **J** UMAP plot depicting 14,767 monocyte-macrophages allocated into the six subsets. **K** Volcano plot highlighting the top 10 genes with the highest and lowest expression levels in each monocyte-macrophage subset compared to others. **L** Expression levels of classical monocyte-macrophage marker genes across subsets, with normalized expression value shown on the y-axis. **M** Box plot illustrating differences in the proportions of monocyte-macrophage subsets among different treatments. Number of samples included in the analysis: PBS (Sed), n = 3; LPS (Sed), n = 3; LPS (Exe), n = 4. Data are represented as mean ± SEM. The lower whisker, lower hinge, box center, upper hinge, and upper whisker represent the minimum, lower quartile, median, upper quartile. Data are represented as mean ± SEM. Statistical analysis was performed using two-tailed unpaired t-tests (**A, B, D, E, G, I,** and **M**). NS, not significant. Scale bars: 100 μm (**F, H**). Source data for **A, B, D, E, G, I,** and **M** are provided in the Source Data file.

in MHC II^high Mø in the septic heart. However, regular physical exercise counteracted this reduction, restoring MHC II^high Mø levels to normal (Fig. 3M). Additionally, distinct dendritic cell (DC) and T cell subpopulations were identified in the cardiac tissue (Figs. S3 and S4), and cell-to-cell communication analysis was performed to understand how exercise modulates the interactions and coregulation within immune cells in septic hearts (Fig. S5). Collectively, these findings suggest that regular exercise modulates the differentiation and composition of monocyte and macrophage subsets, thereby protecting cardiac function in SICM.

## Regular exercise induces a monocyte-derived iNOS⁺ subset to protect septic cardiac function

Considering that the replenishment of Ccr2⁺ Mø and MHC II^high Mø is fully or partially dependent on the differentiation of Ly6c2^high Mo, respectively, we performed pseudotime analyses for trajectory inference using the Monocle2 algorithm. Interestingly, Ly6c2^high Mo was localized at the beginning of the pseudotime trajectory, whereas MHC II^high Mø and iNOS⁺ Mo were at the end (Fig. 4A). We then modeled gene expression along the MHC II^high Mø and iNOS⁺ Mo lineages, and identified five gene sets with specific expression patterns. Analysis of the representative genes and functional enrichment for each pattern revealed that genes related to interleukin-1 production, regulation of T cell activation, and leukocyte migration were gradually upregulated during the differentiation from Ly6c2^high Mo to MHC II^high Mø. During the early stage of Ly6c2^high Mo differentiation and the process towards iNOS⁺ Mo, genes associated with functions such as phagocytosis, actin filament polymerization, and regulation of lymphocyte proliferation were active. Concurrently, genes that regulate anti-inflammatory and pro-reparative functions, such as *Arg1* and *FabpS*, were continuously expressed. Notably, antibacterial-related functions were significantly activated upon differentiation into iNOS⁺ Mo (Fig. 4B), suggesting a frontline role of this cell type in cardiac sepsis.

To elucidate the effects of iNOS⁺ monocytes in SICM, we designed a targeted nano-inhibitor using the small-molecule inhibitor TRIM to specifically inhibit iNOS in Ly6C⁺ monocytes (Fig. 4C). The TRIM was encapsulated within the core of a lipidosome constructed with PEG-DSPE and DSPE-PEG-NHS, with an antibody targeting Ly6C grafted onto the liposome surface for precise monocyte targeting. The targeted nano-inhibitor exhibited a monodisperse particle size distribution with a particle size of 93.12 nm and a regular spherical shape (Fig. S6A–C). The drug loading content and antibody grafting content were calculated to be 38.53% and 2.13%, respectively. Furthermore, the stability of the targeted nano-inhibitor was assessed by monitoring long-term particle size variations, revealing no significant disassembly over time (Fig. S6D). Western blot analysis confirmed that this nanoparticle effectively suppressed iNOS expression in monocytes (Fig. 4D, E). In line with previous findings showing that the lack of inflammatory macrophages in the early phase of cardiac damage leads to the accumulation of apoptotic cardiomyocytes and impaired healing[38–41], we observed that inhibiting monocyte iNOS abolished the beneficial effects of exercise on reducing the drop in body temperature and weight loss during sepsis (Fig. 4F, G). Consistent with our previous observations, physical activity preserved cardiac function (Fig. 4H–J). However, the inhibition of iNOS in monocytes abolished the protective effects of exercise on sepsis-induced myocardial injury. Interestingly, compared to control mice, the inhibition of iNOS in monocytes even exacerbated LPS-induced cardiac dysfunction in sedentary mice. This effect was mirrored by the abolishment of reduced apoptosis and oxidative damage observed in exercised mice (Fig. 4K–P). Further time-course analysis of cardiac macrophages during SICM revealed a shift from iNOS⁺ to Arg1⁺ macrophages (Fig. 4Q–S), with exercise appearing to accelerate this transition (Figs. 1Q–S and 4M, P–S). To further investigate the role of Arg1 in monocytes during SICM, we employed the previously described nanoparticle to specifically inhibit

Arg1 in Ly6C⁺ monocytes (Figs. 4C and S6E–G). Arg1 inhibition in monocytes abolished the protective effects of exercise on reducing body temperature decline, weight loss, IL-1β and TNF-α levels and sepsis-induced myocardial injury (Figs. 4T–X and S6H–K). Additionally, the previously observed accelerated transition of cardiac macrophages from iNOS⁺ to Arg1⁺ macrophages was also disrupted by Arg1 inhibition in monocytes (Fig. S6L). In summary, these findings indicate that the expression of both iNOS and Arg1 in monocyte-derived macrophages contributes to the protection of cardiac function during SICM, and exercise accelerates the transition from iNOS⁺ to Arg1⁺ macrophages.

## Exercise boosts glycolysis and subsequent histone lactylation in monocyte-derived macrophages during SICM

We then aimed to elucidate the mechanisms by which iNOS⁺ monocytes exhibit anti-inflammatory functions, especially whether and how exercise could influence their metabolic characteristics. Previous findings from us and other labs have demonstrated that metabolic alterations in macrophages can affect their functions[15,16,42]. Firstly, we utilized the Compass algorithm for flux balance analysis to evaluate the activity of specific reactions within genome-scale metabolic networks. The Compass-score differential activity test revealed significantly elevated metabolic fluxes in sedentary SICM hearts. In contrast, exercise rewired macrophage metabolism by promoting or inhibiting various metabolic pathways in septic hearts (Fig. S7). To delve deeper, we used scMetabolism analysis to assess how exercise modulates the metabolic profile in total and specific subsets of cardiac monocytes and macrophages (Fig. S8). Our findings revealed that exercise markedly enhanced glycolysis while suppressing the TCA cycle in the entire monocyte and macrophage populations, monocytes, macrophages and iNOS⁺ Mo compared to those from sedentary SICM hearts (Figs. 5A and S8). These observations were consistent with the Compass-score analysis (Figs. 5B and S7).

To validate these findings, we isolated monocytes from peripheral blood and performed an extracellular acidification rate (ECAR) analysis. Glycolysis was approximately 55% higher in monocytes from exercised septic mice compared to sedentary counterparts (Fig. 5C, D). This increase in anaerobic glycolysis was corroborated by elevated cellular lactate levels, which were about 50% higher in exercised monocytes than in sedentary controls following LPS injection (Fig. 5E). Recent studies have shown that glycolysis-derived lactate can modulate histone lactylation, thereby triggering wound-healing gene expression[43,44]. Notably, we observed exercise-induced global histone lysine lactylation (Kla) in both circulating monocytes and cardiac macrophages in SICM (Fig. 5F–I). Collectively, these findings suggest that exercise enhances glycolysis in monocytes, leading to increased histone lactylation in monocyte-derived macrophages, and this metabolic reprogramming potentially contributes to the protection of cardiac function in SICM.

## Voluntary running induces histone lactylation in monocyte-derived macrophages to protect cardiac function in SICM

To assess the clinical relevance of this protection mechanism, we isolated peripheral blood monocytes from both SICM patients and healthy donors using CD14⁺ beads and performed RNA-seq analysis. In line with the observations in mice, monocytes from SICM patients exhibited increased glycolysis and TCA cycle compared to those from healthy donors (Fig. 6A and Table S2). Interestingly, the expression of lactate dehydrogenase A (LDHA), a crucial enzyme in anaerobic glycolysis that converts pyruvate to lactate, was significantly elevated in SICM patient monocytes (Fig. 6B). To further investigate the role of exercise-induced histone lactylation in monocyte-derived macrophages in mitigating SICM, we generated mice lacking *Ldha* specifically in myeloid cells. This was achieved by intercrossing *Ldha* floxed mice (*Ldha* ^L/L) with the myeloid cell-specific LysM: Cre transgenic line

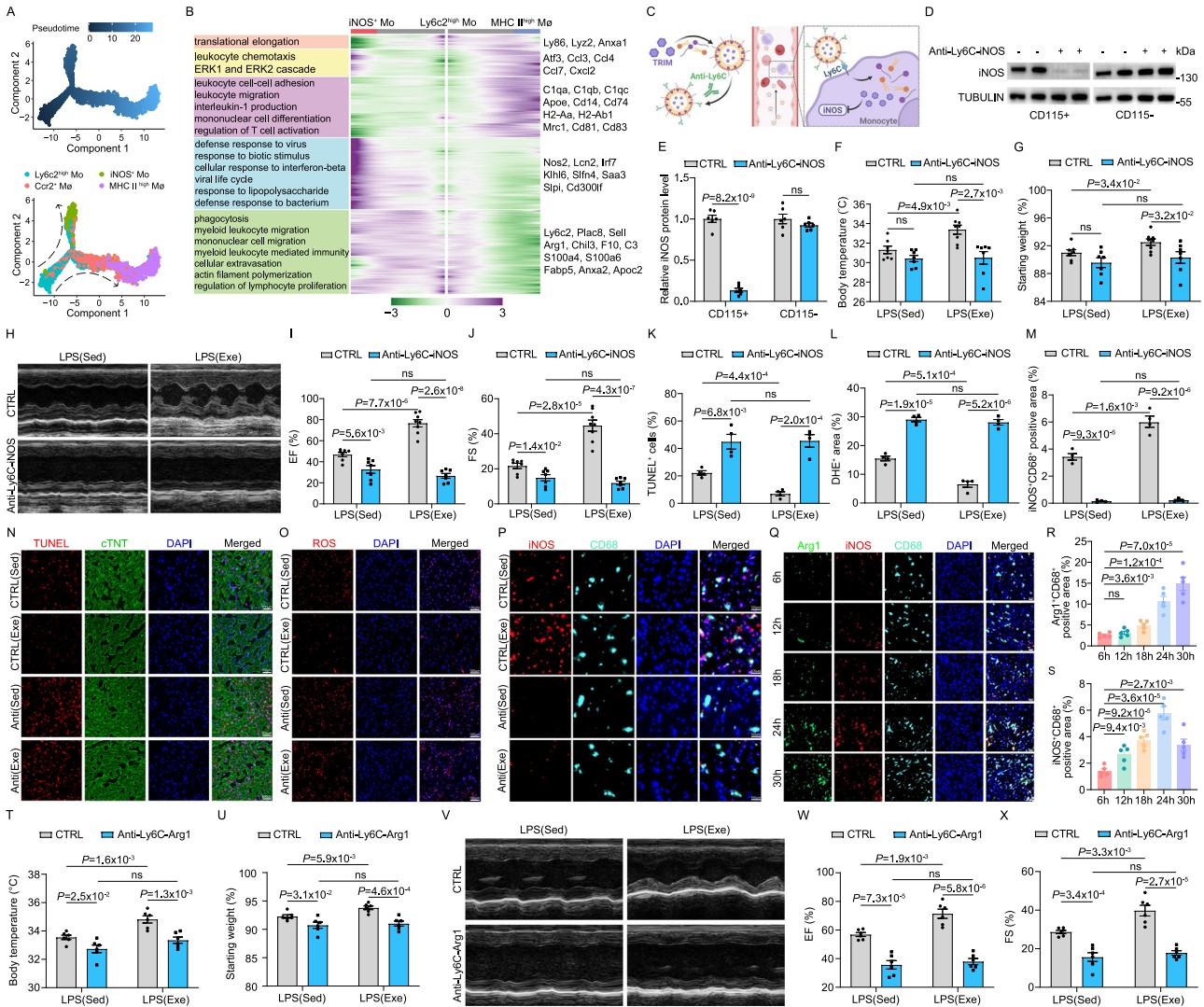

**Fig. 4 | Regular exercise induces a monocyte-derived iNOS⁺ subset to protect septic cardiac function. A** The Monocle prediction of the monocyte-macrophage developmental trajectory for Ly6c2$^{high}$ Mo, Ccr2$^+$ Mø, MHC II$^{high}$ Mø and iNOS$^+$ Mo. **B** Heatmap displaying differentially expressed genes based on their shared kinetics through pseudotime using Monocle2. Distinct GO terms and representative genes associated with each model are shown on the left and right. **C** Schematic illustration of the nano-inhibitor designed to specifically inhibit iNOS in monocytes. Figure created in BioRender, Shang, M. (2025) lbdglen. **D, E** Efficiency of iNOS inhibition by the nano-inhibitor in monocytes, detected by WB. β-Tubulin was used as loading control. Numbers represent densitometric fold change relative to β-Tubulin. The analysis was performed on 6 samples per group. Body temperature (**F**) and percent change in body weight (**G**) measured 18 h after i.p. injection of LPS with Anti-Ly6C-iNOS or control in sedentary (Sed; CTRL, n = 7; Anti-Ly6C-iNOS, n = 7) and exercise (Exe; CTRL, n = 8; Anti-Ly6C-iNOS, n = 7) mice. Representative echocardiography images (**H**) and quantification of EF% (**I**), FS % (**J**) at 18 h post-LPS i.p injection with Anti-Ly6C-iNOS or control in Sed (CTRL, n = 7; Anti-Ly6C-iNOS, n = 7) and Exe

(CTRL, n = 8; Anti-Ly6C-iNOS, n = 7) mice. Representative images and quantification of TUNEL⁺ cells (**N, K**), DHE⁺ areas (**O, L**) and iNOS⁺ CD68⁺ cells (**P, M**) in hearts of Sed and Exe mice 18 h post LPS injection, treated with either Anti-Ly6C-iNOS or control (CTRL). The analysis was performed on 4 samples per group from 2 independent experiments. **Q–S** Representative images and quantification of Arg1, CD68 and iNOS in hearts of mice over a time course following LPS injection. The analysis was performed on 5 samples per group from 3 independent experiments. Body temperature (**T**) and percent change in body weight (**U**) measured 18 h after i.p. injection of LPS with Anti-Ly6C-Arg1 or control in Sed and Exe mice. The analysis was performed on 6 samples per group. Representative echocardiography images (**V**) and quantification of EF% (**W**) and FS% (**X**) measured 18 h after i.p. injection of LPS with Anti-Ly6C-Arg1 or control in Sed and Exe mice. The analysis was performed on 6 samples per group. Data are represented as mean ± SEM. Statistical analysis was performed using two-tailed unpaired t-tests. NS not significant. Scale bar: 100 μm (**N–P**), 50 μm (**Q**). Source data for **D–G, I–M, R–U** and **W–X** are provided in the Source Data file.

(Fig. S9A, B), hereafter referred to as *Ldha*$^{ΔMo}$ mice. These *Ldha*$^{ΔMo}$ mice selectively inhibit histone lactylation in monocytes and abolish exercise-induced increase in histone Kla (Fig. S9C).

　　Similar to the effects of iNOS or Arg1 inhibition in monocytes, *Ldha*$^{ΔMo}$ mice exhibited a reversal of the protective effects of regular exercise on body temperature drop and body weight loss during sepsis (Fig. 6C, D). Consistent with these observations, *Ldha*$^{ΔMo}$ mice also showed a loss of the improved septic cardiac function and reduced apoptosis and oxidative damage in septic hearts that were typically induced by regular physical exercise (Fig. 6E–G). Furthermore,

immunofluorescence analysis aligned with our previous observations, demonstrating that exercise reduced cardiomyocyte apoptosis and oxidative damage in septic hearts, which were dismissed in *Ldha*$^{ΔMo}$ mice (Fig. 6H–K). Intriguingly, exercise increased histone lactylation in cardiac macrophages by nearly threefold compared to sedentary controls. This increase was associated with the expression of the wound-healing marker Arg1, as histone lactylation is crucial for Arg1 expression in exercised SICM mice hearts, with approximately 70% of Arg1⁺ macrophages exhibiting histone lactylation. However, both the enhanced histone lactylation and Arg1 expression were abolished in

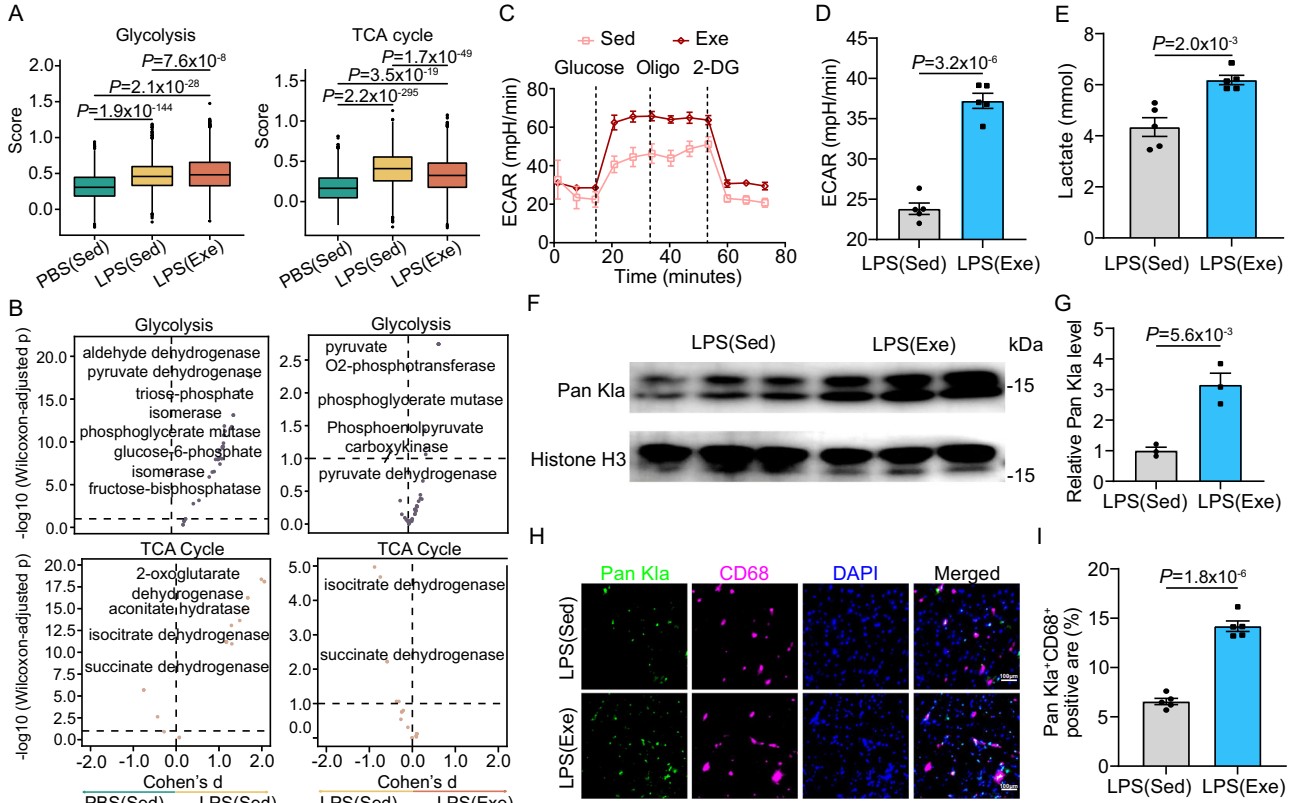

**Fig. 5 | Exercise boosts glycolysis and subsequent histone lactylation in monocyte-derived macrophages during SICM. A** Activity scores of glycolysis and TCA cycle assessed using single-cell metabolism analysis (scMetabolism). Number of cells included in the analysis in sedentary (Sed) or exercise (Exe) group: PBS (Sed), n = 2722; LPS (Sed), n = 2517; LPS (Exe), n = 7940. The lower whisker, lower hinge, box center, upper hinge, and upper whisker represent the minimum, lower quartile, median, upper quartile. **B** Compass-score differential activity test showing the metabolic flux levels of glycolysis and the TCA cycle. **C**, **D** Extracellular acidification rate (ECAR) in monocytes isolated from Exe or Sed mice 18 h post LPS injection. The graph shows values from 5 biological repetitions per condition. **E** Levels of intracellular L-lactate in monocytes from mice 18 h post LPS injection.

The analysis was performed on 5 samples per group. **F**, **G** Western blot for Pan Kla in monocytes from mice 18 h post-injection with PBS or LPS. Histone H3 was used as an internal control, with densitometric fold change relative to Histone H3 indicated. The analysis was performed on 3 samples per group. **H**, **I** Representative images and quantification of CD68 and Pan Kla in SICM hearts of mice with or without exercise. The analysis was performed on 5 samples per group. All experiments except A and B show representative values from three independent experiments. Data are represented as mean ± SEM. Statistical analysis was performed using two-tailed unpaired t-tests (**A**, **D**, **E**, **G**, **I**). Scale bar: 100 μm (**H**). Source data for **A**, **C**–**E**, **G**, and **I** are provided in the Source Data file.

$Ldha^{\Delta Mo}$ mice (Figs. 6L–P and S9D). Overall, these data suggested that exercise-induced histone lactylation in monocytes and monocyte-derived macrophages plays a pivotal role in preserving cardiac function during sepsis.

### Exercise enhanced H3K18 histone lactylation in monocytes from both humans and mice

Next, we sought to identify the specific histone lactylation sites induced by exercise in monocytes. In addition to our previous observation that exercise enhanced Pan Kla levels during SICM (Fig. 5F, G), we found that exercise also elevated Pan Kla levels in PBS-injected mice compared to sedentary controls (Fig. 7A, B). To identify specific histone lactylation sites, we analyzed lactylation of histone H3 at lysine residue 18 (H3K18la), as well as H3K9la, H3K14la, H4K5la, H4K8la, H4K12la, and H4K16la in monocytes from sedentary and exercise mice subjected to LPS or PBS as controls. H3K18la levels were significantly elevated after exercise compared to sedentary mice with PBS injection as a control and were further increased in LPS-challenged mice (Fig. 7A–C). In contrast, H3K9la, H3K14la, H4K5la, H4K8la, H4K12la, and H4K16la showed no significant changes in response to exercise under either LPS challenge or PBS control conditions (Fig. 7A, D–I). Among these marks, only H3K18la exhibited changes closely mirroring the Pan Kla pattern, indicating that exercise specifically induced H3K18la. These observations raised the question of whether regular

exercise similarly promotes H3K18la in human monocytes. To address this, we recruited volunteers with sedentary and active lifestyles (Fig. 7J and Table S3). Given that lactate levels influence histone lactylation, as reported in previous studies[40,45], we exposed circulating monocytes from mice to varying concentrations of lactate to induce histone lactylation and identified 20 mM lactate as the optimal concentration (Fig. S9E, F). Next, we treated monocytes from sedentary volunteers with 20 mM exogenous lactate and observed increased levels of Pan Kla and H3K18la (Fig. 7K–M). Consistently, monocytes from active volunteers exhibited significantly higher levels of both Pan Kla and H3K18la compared to those from sedentary individuals (Fig. 7N–P).

Next, we investigated the duration of exercise required to induce histone lactylation in monocytes and its protective effects on cardiac function during SICM. A time-course analysis revealed that Pan Kla and H3K18la levels remained comparable in monocytes from mice subjected to 1 or 7 days of exercise but significantly increased after 14 days, with further elevations observed after 28 days of exercise (Fig. S10A–C). In line with these observations, 14 days of exercise mitigated body weight loss and preserved cardiac function upon LPS challenge, while prolonged exercise for 28 days further enhanced these protective effects (Fig. S10D–G). In summary, these findings demonstrate that exercise induces H3K18la in monocytes from both humans and mice, underscoring its pivotal role in preserving cardiac function during SICM.

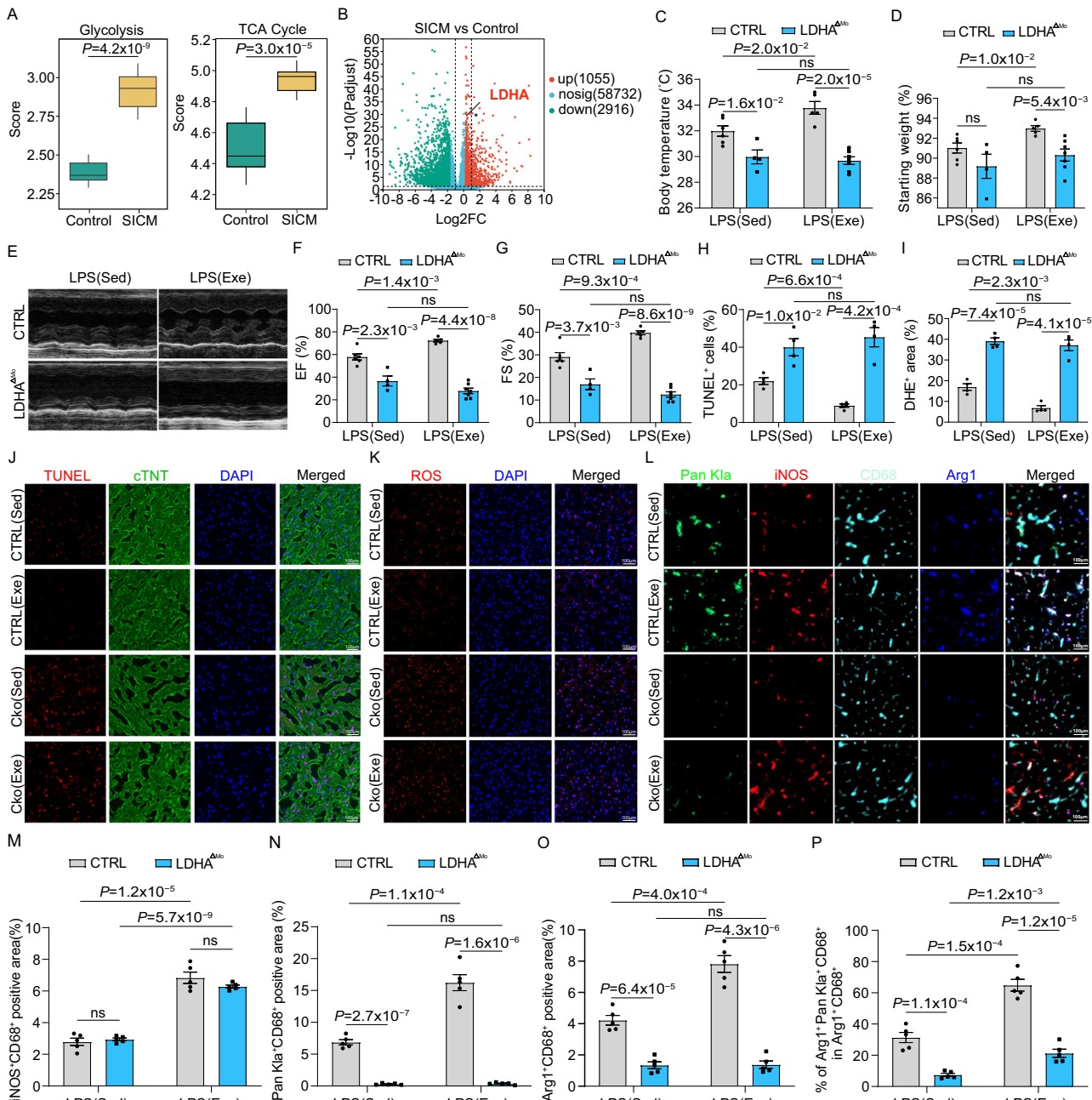

**Fig. 6 | Voluntary running induces histone lactylation in monocyte-derived macrophages to protect cardiac function in SICM. A** Glycolysis and TCA cycle activity scores in peripheral blood monocytes from healthy donors and SICM patients based on GSVA analysis. Number of samples included in the analysis: Control, n = 9; SICM, n = 10. The lower whisker, lower hinge, box center, upper hinge, and upper whisker represent the minimum, lower quartile, median, upper quartile. **B** Volcano plot showing differential gene expression between healthy donors with SICM patients, and LdhA is indicated in the graph. Body temperature (**C**) and percentage change in body weight (**D**) of myeloid specific *Ldha* deletion (*Ldha*^ΔMo) and control mice 18 h post LPS injection, with or without exercise. Groups include sedentary (Sed; CTRL, n = 6; *Ldha*^ΔMo, n = 4) and exercise (Exe; CTRL, n = 5; *Ldha*^ΔMo, n = 7). Representative echocardiography images (**E**) and quantification of EF% (**F**) and FS % (**G**) in *Ldha*^ΔMo or control mice 18 h post-LPS injection, with or

without exercise. Groups include sedentary (CTRL, n = 6; *Ldha*^ΔMo, n = 4) and exercise (CTRL, n = 5; *Ldha*^ΔMo, n = 7). Representative images and quantification of TUNEL (**J, H**) and DHE (**K, I**) positive area in hearts of *Ldha*^ΔMo or control mice 18 h post LPS injection, with or without exercise. The analysis was performed on 4 samples per group. **L–P**, Representative images and quantification of Pan Kla, iNOS, CD68 and Arg1 in the hearts of *Ldha*^ΔMo or control mice 18 h post LPS injection, with or without exercise. The analysis was performed on 5 samples per group. All experiments except A and B show representative values from three independent experiments. Data are represented as mean ± SEM. Statistical analysis was performed using two-tailed unpaired t-tests, except for A, where unpaired Mann–Whitney was used. NS not significant. Scale bar: 100 μm (**J–L**). Source data for **A–D**, **F–I** and **M–P** are provided in the Source Data file.

## Exercise enhanced H3K18 histone lactylation in monocytes via p300 and HDAC2

Next, we aimed to identify the lactyltransferase and delactylase for the addition or removal of exercise-induced H3K18la in monocytes. P300 has been reported as a potential writer for histone lactylation[43,46]. To

validate its role in exercise-induced lactylation, we silenced p300 in monocytes using siRNA and subsequently incubated the cells with lactate to induce lactylation, as previously described[44]. Consistently, lactate-induced histone lactylation was observed at both global Pan Kla and H3K18la levels (Fig. 8A–C). However, this induction was abrogated

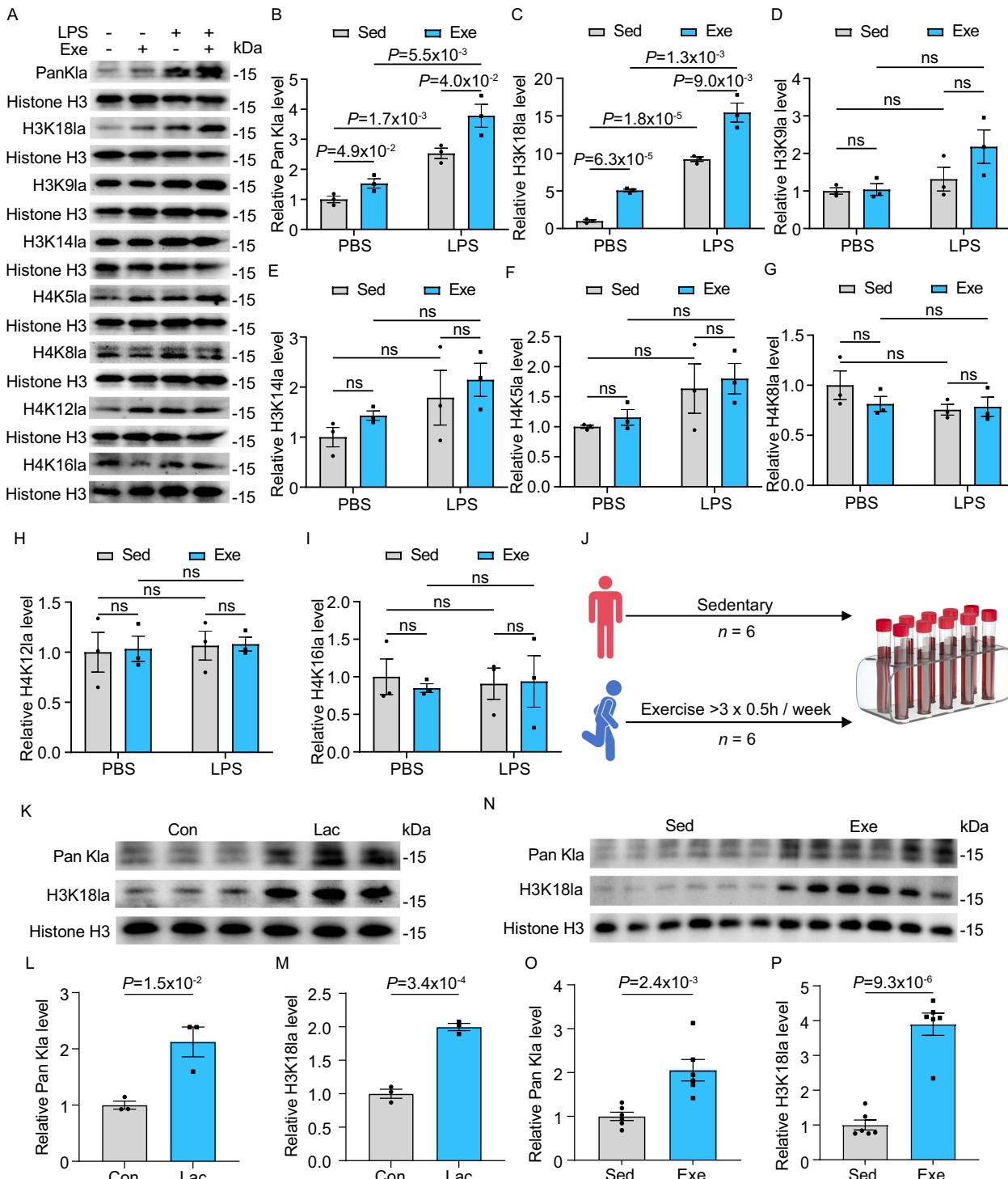

**Fig. 7 | Exercise enhanced H3K18 histone lactylation in monocytes from both humans and mice.** **A–I** Western blot for detecting lactylation sites, including Pan Kla, H3K18la, H3K9la, H3K14la, H4K5la, H4K8la, H4K12la, and H4K16la in monocytes from exercise (Exe) and sedentary (Sed) mice 18 h post injection with PBS or LPS. Histone H3 was used as an internal control, with densitometric fold change relative to Histone H3 indicated. The analysis was performed on 3 samples per group of 3 independent experiments. **J** Schematic diagram for recruiting volunteers with a sedentary and active lifestyle. **K–M** Western blot for Pan Kla and H3K18la in human monocytes incubated with 20 mM sodium lactate for 24 h. Histone H3 was

used as an internal control, with densitometric fold change relative to Histone H3 indicated. The analysis was performed on 3 samples per group of 2 independent experiments. **N–P**, Western blot analysis for Pan Kla and H3K18la in human monocytes, including individuals with regular exercise habits and sedentary controls. Histone H3 was used as an internal control, with densitometric fold change relative to Histone H3 indicated. The analysis was performed on 6 samples per group. Data are represented as mean ± SEM. Statistical analysis was performed using two-tailed unpaired t-tests. NS, not significant. Source data for **B–I** and **L–P** are provided in the Source Data file.

when p300 was silenced (Fig. 8D–F). These findings indicate that p300 functions as the histone lactylation writer for H3K18la in monocytes. To identify the delactylase responsible for histone lactylation removal, we focused on HDAC1-3, which have been reported as potential erasers of histone lactylation[47]. Using siRNA, we silenced HDAC1, HDAC2, and HDAC3 and assessed lactate-induced H3K18la levels in monocytes. Knockdown of HDAC2 resulted in a marked increase in H3K18la levels (Fig. 8G–I), whereas silencing HDAC1 or HDAC3 did not elevate H3K18la levels (Fig. S11A–F). These results identify HDAC2 as the principal delactylase of H3K18la in monocytes. In summary, these results indicate that p300 and HDAC2 function as the lactyltransferase and delactylase, respectively, for exercise-induced H3K18la in monocytes, contributing to its protective effects in SICM.

To elucidate whether H3K18la regulates the previously observed accelerated transition from iNOS$^+$ to Arg1$^+$ macrophages, we isolated bone marrow-derived macrophages (BMDM) and incubated them with lactate for 24 h. Following this preconditioning, the BMDMs were challenged with LPS in a time-course experiment. Lactate preconditioning significantly enhanced Arg1 expression without affecting iNOS expression, thereby facilitating a more rapid transition to wound-healing macrophages (Fig. 8J–O). This suggests that lactate-preconditioned macrophages contribute to accelerated restoration of immune homeostasis in response to inflammatory challenges. To investigate how H3K18la regulates *Arg1* expression, we performed genome-wide Cleavage Under Targets and Tagmentation (CUT&Tag) analysis using an anti-H3K18la antibody, followed by high-throughput DNA sequencing. The data revealed significant enrichment of H3K18la in the promoter regions of *Arg1* (Fig. 8P). Additionally, CUT&Tag-qPCR analysis indicated increased H3K18la enrichment in the promoter regions of *Arg1* following LPS stimulation compared to PBS injection, and to a higher extent in the context of regular exercise (Fig. 8Q). In summary, these results demonstrate that exercise-induced H3K18la in monocytes, regulated by p300 and HDAC2, plays a critical role in the protective effects of exercise in SICM by promoting a faster transition to wound-healing macrophages and restoring immune homeostasis.

### Lactate-educated monocyte transfusion improves cardiac function in SICM

We finally assessed the therapeutic implications of our findings. To this end, we performed several experiments. First, we utilized a more clinically relevant model, the cecal ligation and puncture (CLP) model, to induce SICM in polymicrobial sepsis. Consistent with the observations in the LPS-induced model, exercise significantly prevented the drop in body temperature, weight loss, and elevated IL-1β and TNF-α production in CLP-induced SICM (Fig. S12A–D). Additionally, exercised mice showed significantly improved cardiac function, reduced apoptosis, and mitigated oxidative stress in the hearts of SICM mice (Fig. S12E–K). Next, we performed adoptive transfer experiments in which monocytes isolated from exercised donor mice were transfused into sedentary recipients post-LPS injection to induce sepsis. Monocytes from exercised mice exhibited increased glycolysis compared to those from sedentary mice (Fig. S12L–N). Monocyte transfusion significantly alleviated the drop in body temperature and body weight, as well as reduced IL-1β and TNF-α production caused by sepsis (Fig. 9A–D). Furthermore, transfusion of monocytes from exercised mice significantly improved cardiac function and reduced apoptosis in the hearts of SICM mice (Fig. 9E–I). In summary, these results demonstrate that training of circulating monocytes by exercise plays a central role in protecting against SICM.

Next, to mimic the benefits of voluntary exercise, we educated circulating monocytes with lactate to induce histone lactylation (Fig. S9E, F). These lactate-educated monocytes were labeled with PKH26 and transfused into SICM mice via tail vein injection. The labeled monocytes were confirmed to localize in the heart of SICM mice (Fig. 9J, K). Notably, similar to the protective effects observed

with voluntary exercise, transfusion of monocytes with elevated histone lactylation significantly enhanced cardiac function and reduced apoptosis and oxidative damage in septic hearts (Fig. 9L–T). To further confirm the protective role of histone lactylation against SICM, we assessed the in vivo impact of p300-inhibited monocytes. Monocytes were pretreated with C646, a selective inhibitor of p300[48], before adoptive transfer into SICM mice. Mice receiving p300-inhibited monocytes exhibited an abrogation of the protective effects of exercise, including loss of improvements in body temperature, weight loss, cardiac function and myocardial apoptosis (Fig. S13A–J). These findings indicate that H3K18la, with p300 as the writer, is essential for preserving cardiac function in SICM. In summary, our results indicate that, akin to the benefits of voluntary exercise, lactate-induced histone lactylation in monocytes protects cardiac function by restoring immune homeostasis in SICM.

## Discussion

The cardiac immune environment plays a critical role in maintaining myocardial homeostasis and facilitating repair after injury[49–52]. Macrophages, as the most abundant and functionally diverse immune cells in the heart, significantly contribute to the repair of damaged myocardium following cardiac injury[53]. Upon injury, circulating monocytes are recruited to the myocardium, where they differentiate into macrophages. During the early phase of cardiac damage, monocyte-derived macrophages are essential for clearing tissue debris and releasing cytokines, growth factors, and fibroblast growth factors[54,55]. The importance of inflammatory cells in the acute phase of cardiac injury is further supported by evidence showing that macrophage depletion impairs the clearance of tissue debris and dead cells, including necrotic cardiomyocytes[40,41]. As the repair process progresses, macrophages enhance the expression of reparative genes, remodeling fibroblasts, promoting angiogenesis, and contributing to extracellular matrix (ECM) degradation to restore tissue integrity[56,57]. However, how an active lifestyle modulates the cardiac immune environment has not been fully elucidated.

In this study, we investigated the molecular mechanisms underlying exercise-induced myocardial protection in SICM using bulk RNA-seq and scRNA-seq. Bulk RNA-seq analysis revealed that exercise primarily protects the myocardium by suppressing the excessive inflammation associated with SICM. Building on these observations, we utilized scRNA-seq to analyze immune cells from SICM hearts subjected to exercise, thereby constructing a detailed cardiac immune cell landscape. Among these cells, we identified an exercise-induced iNOS$^+$ Arg1$^+$ monocyte-derived macrophage subset. This subset is characterized by the expression of both inflammatory and wound-healing genes, suggesting a dual role in the cardiac immune response during SICM. The presence of these iNOS$^+$ Arg1$^+$ subsets suggests a dual role in the cardiac immune response during SICM. Initially, they appear to enhance pro-inflammatory immune responses, which are crucial for combating early-stage infections. Subsequently, they facilitate the transition of cardiac macrophages from an inflammatory to a wound-healing state, thereby promoting the restoration of cardiac homeostasis. This dual functionality underscores the potential of exercise to modulate immune responses in a manner that supports both immediate defense mechanisms and long-term tissue repair and recovery.

It is widely acknowledged that macrophages play a critical role in both promoting inflammation and wound healing processes[58–60]. However, the delicate balance between these two paradigms and how they can be harnessed to improve heart repair and functional recovery following injury remains relatively underexplored. In this study, we demonstrate that exercise-induced lactate preadapts inflammatory macrophages to fine-tune their resolution phase after fulfilling their pro-inflammatory function of clearing cellular debris. Exercise reprograms monocyte metabolism to enhance glycolysis,

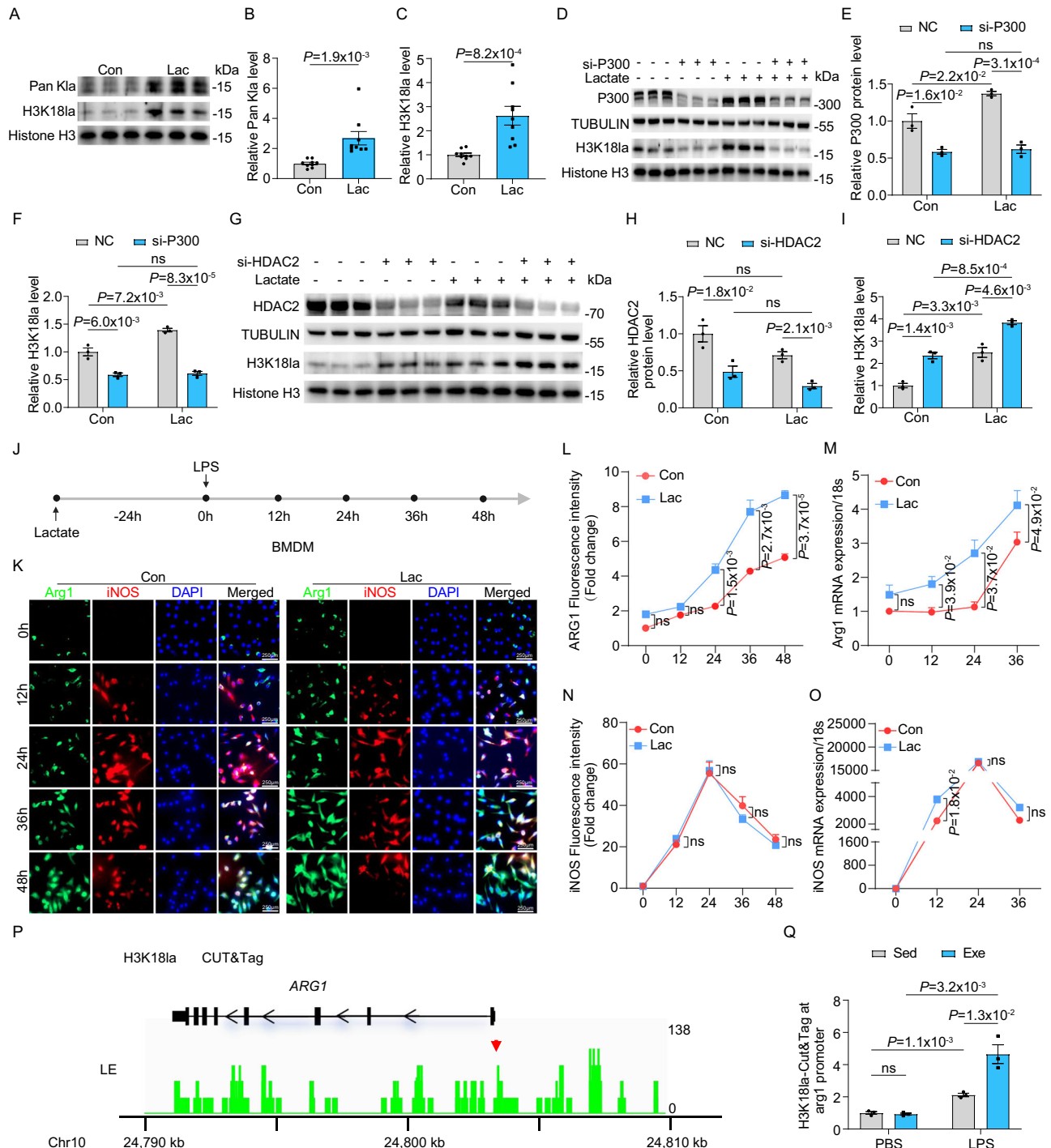

**Fig. 8 | Exercise enhanced H3K18 histone lactylation in monocytes via p300 and HDAC2. A–C** Western blot for Pan Kla and H3K18la in mice monocytes incubated with 20 mM sodium lactate for 24 h. Histone H3 was used as an internal control, with densitometric fold change relative to Histone H3 indicated. The analysis was performed on 9 samples per group. Western blot for H3K18la in mouse monocytes subjected to p300 or HDAC2 knockdown using si-P300 (**D–F**) or si-HDAC2 (**G–I**), treatment with 20 mM sodium lactate for 24 h, and their respective control groups. Tubulin and Histone H3 were used as internal controls, with densitometric fold change relative to Tubulin and Histone H3 indicated. The analysis was performed on 3 samples per group. **J** Schematic presentation of BMDMs stimulated with LPS over a time course following incubation with 20 mM sodium lactate for 24 h. **K, L, N** Representative images and quantification of Arg1 and iNOS

in lactate pre-incubated BMDMs upon LPS challenge. The analysis was performed on 4 samples per group. RT-qPCR analysis of *Arg1* (**M**) and *iNOS* (**O**) in lactate pre-incubated BMDMs upon LPS challenge. The analysis was performed on 3 samples per group. Representative traces of CUT&Tag analysis showed that H3K18la was enriched in the promoter regions of *Arg1* (**P**). CUT&Tag results for *Arg1* were verified by qPCR (**Q**). The analysis was performed on 3 samples per group. Exercise is abbreviated as Exe and sedentary as Sed. All experiments show representative values from at least 2 independent experiments. Data are represented as mean ± SEM. Statistical analysis was performed using two-tailed unpaired t-tests. NS, not significant. Scale bar: 250 µm (**K**). Source data for **B**, **C**, **E**, **F**, **H**, **I**, **L−O** and **Q** are provided in the Source Data file.

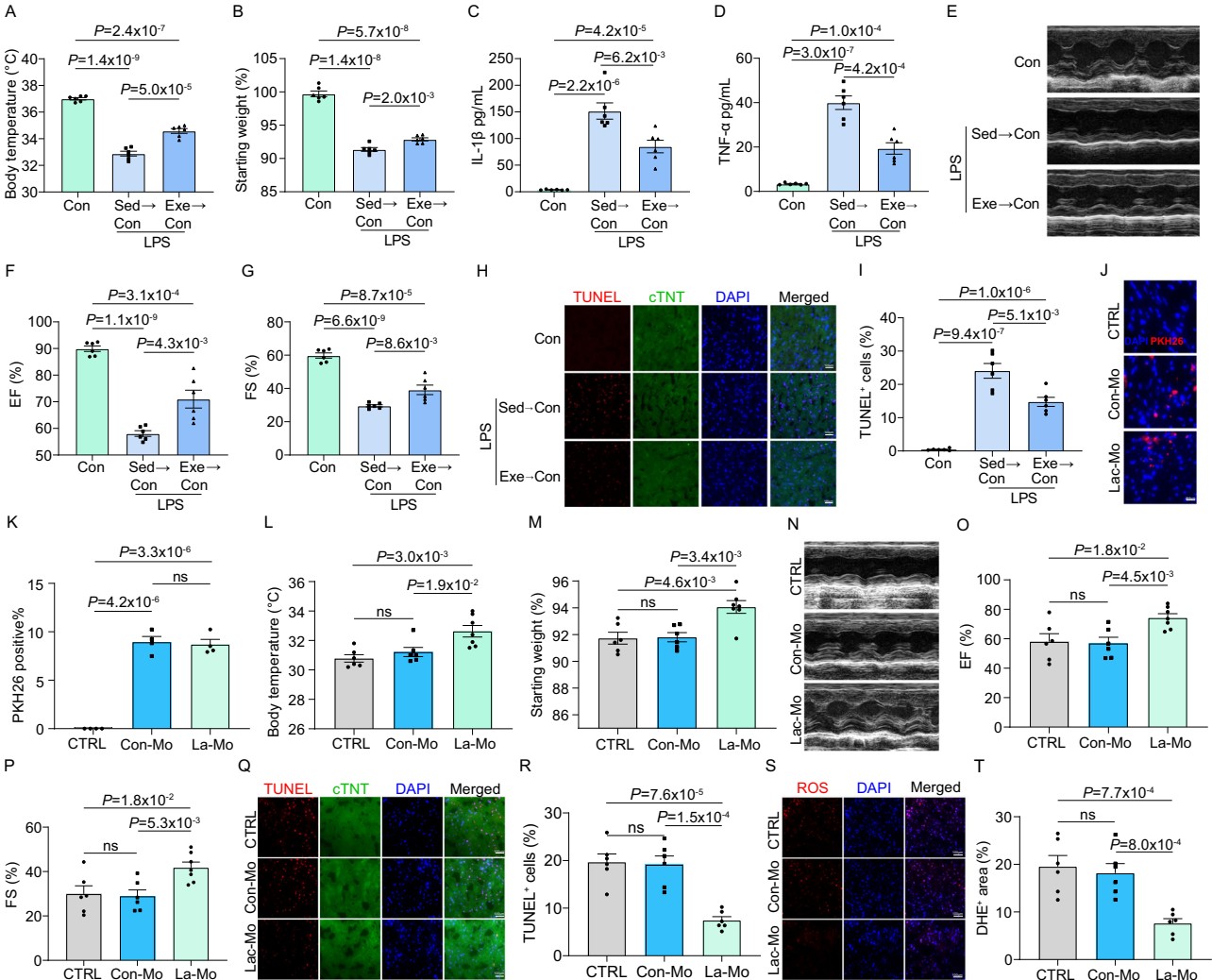

**Fig. 9 | Lactate-educated monocyte transfusion improves cardiac function in SICM.** Body temperature (**A**) and percent change in body weight (**B**) were measured in SICM mice following reinfusion with monocytes from sedentary (Sed) or exercised (Exe) mice. The analysis was performed on 6 samples per group. ELISA analysis of IL-1β (**C**) and TNF-α (**D**) in SICM mice following reinfusion with monocytes from sedentary or exercised mice. The analysis was performed on 6 samples per group. Representative echocardiography images (**E**) and quantification of EF% (**F**) and FS% (**G**) in SICM mice following reinfusion with monocytes from sedentary or exercised mice. The analysis was performed on 6 samples per group. Representative images (**H**) and quantification (**I**) of TUNEL in heart sections from SICM mice following reinfusion with monocytes from sedentary or exercised mice. The analysis was performed on 6 samples per group. **J, K** Representative images and quantification of PKH26 labeled monocytes infiltration into SICM hearts 18 h post monocyte reinfusion. The analysis was performed on 4 samples per group of 2 independent experiments. Body temperature (**L**) and percentage change in body weight (**M**) in SICM mice following reinfusion with lactate-educated monocytes. Number of samples included in the analysis: n = 6 for CTRL, n = 6 for Con-Mo, n = 7 for La-Mo. Representative echocardiography images (**N**) and quantification of EF% (**O**) and FS% (**P**) in SICM mice after lactate-educated monocyte reinfusion. Number of samples included in the analysis: n = 6 for CTRL, n = 6 for Con-Mo, n = 7 for La-Mo. Representative images and quantification of TUNEL (**Q, R**) and DHE (**S, T**) positive areas in hearts of SICM mice post lactate-educated monocyte reinfusion. The analysis was performed on 6 samples per group. Data are represented as mean ± SEM. Unpaired two-tailed t-test was used to determine the statistical significance. NS not significant. Scale bar: 100 μm (**H, Q, S**), 200 μm (**J**). Source data for **A−D, F, G, I, K−M, O, P, R** and **T** are provided in the Source Data file.

and the subsequently produced lactate activates the endogenous "lactate clock" by boosting histone lactylation, which in turn induces the expression of reparative genes[43].

Initially, by specifically inhibiting iNOS in monocytes, we found that iNOS⁺ pro-inflammatory macrophages are essential for cell debris clearance upon LPS challenge. Furthermore, we sought to understand what drives this monocyte-derived macrophage subset to express homeostatic genes and accelerate the resolution process, thereby avoiding overwhelming inflammation, which can lead to organ damage and even organ failure. It has long been known that exercise induces lactate production[61]. Here in our study, we proved that exercise boosts monocyte glycolysis, thereby increasing lactate production. Once produced, the increased lactate levels drive the epigenetic modification of histone lactylation, enhancing the expression of reparative

genes, including *Arg1*. Along with recent studies, we describe a metabolic reprogramming-driven epigenetic modification of monocytes upon exercise, which subsequently infiltrates into the injured heart to accelerate the resolution of cardiac overactivated inflammation. In this study, we identify the existence and the physiological relevance of a metabolite-based macrophage-educated mode, whereby macrophages preadapt their response to immune challenges, accelerating inflammation resolution, and thereby maintaining cardiac function and promoting myocardial recovery (Fig. S14).

Beyond the mechanistic insights, our study holds significant social and clinical implications. Understanding how an active versus sedentary lifestyle impacts cardiac health and disease progression offers a cost-effective and efficient strategy for addressing major global health challenges, including CVD and SICM[23,29]. Beyond exercise-induced

histone lactylation, transfusing monocytes with enhanced lactylation may be considered a therapeutic strategy for CVD, particularly in SICM, by restoring cardiac immune homeostasis and improving cardiac function. Despite these promising advancements, certain limitations of this study warrant further exploration. First, we evaluated the role of exercise and monocyte-derived macrophages in cardiac function only 18 h after LPS injection. Future studies incorporating additional time points following LPS stimulation could provide valuable insights into their role in long-term cardiac remodeling. Second, while we demonstrated that monocytes from active volunteers exhibited significantly higher levels of both Pan Kla and H3K18la compared to the sedentary individuals, these findings align with conclusions reached in the mouse study. However, technical limitations prevented us from thoroughly evaluating the expression of Pan Kla and H3K18la in monocytes from SICM patients. This limitation restricted our ability to establish direct correlations between histone lactylation levels and cardiac function outcomes in SICM patients, which would have strengthened the translational relevance of our findings. Future studies are essential to investigate histone lactylation in monocytes from SICM patients and to correlate these expression levels with clinical outcomes, such as cardiac function progression and mortality rates.

In our previous studies on muscle regeneration[16], we demonstrated that reprogramming macrophage metabolism supports the muscle stem cell pool and promotes their differentiation. In this study, we show that exercise orchestrates monocyte metabolism, enhancing histone lactylation and preadapting monocyte-derived macrophages to accelerate their transition from inflammatory to reparative macrophages, thereby maintaining cardiac function during SICM. In this study, we first illustrate the cardiac immune landscape shaped by regular exercise. Secondly, we demonstrate that histone lactylation induced by enhanced glycolysis in monocytes from exercised mice protects cardiac functions in SICM. Lastly, our data suggest that monocyte adoptive therapy, which mimics the beneficial effects of regular physical activity, represents a promising therapeutic strategy for protecting cardiac function during SICM. These findings open up exciting new possibilities and pave the way for future research and novel treatments for cardiovascular diseases through metabolic and epigenetic reprogramming.

## Methods

### Animals

Male C57BL/6 wild-type mice (Strain No. N000013) were purchased from GemPharmatech (Nanjing, China). A floxed LDHA (MGI: 96759) transgenic mouse line on a C57BL/6 background was intercrossed with a myeloid cell deleter LysM:Cre transgenic line to generate $Ldha^{L/L}$ x LysM:Cre transgenic mice. Floxed LDHA mice (Strain No. T007813) and LysM:Cre mice (Strain No. T003822) were purchased from Gem-Pharmatech (Nanjing, China). Littermates negative for the LysM-Cre transgene ($LdhA^{fl/fl}$) were used as controls. Male mice used in sepsis experiments were aged between 8 and 10 weeks. All mice were housed under standardized feeding and management conditions in specific pathogen-free (SPF) barrier facilities. Environmental parameters were carefully controlled, including a temperature of $22 \pm 1\,°C$, humidity of $50\% \pm 5\%$, and a circadian rhythm system maintaining a 12-h light/12-h dark cycle. Housing and all experimental animal procedures were approved by the Animal Care and Use Committee of the School of Medicine, Zhejiang University.

### Voluntary wheel running training

Male mice were randomly assigned to either a sedentary group (Sed) or an exercise (Exe) group. Mice in the Exe group were individually housed in cages equipped with wheels, allowing for voluntary exercise over a period of three months. The running wheels were connected to cycle computers (SunDing SD-568AE) to monitor activity levels, as previously described[16]. In contrast, the sedentary control mice were housed in the same cages with locked wheels for the same time.

### LPS-induced sepsis model

Following a 3-month voluntary exercise, mice were randomly assigned to receive an i.p. injection of either LPS or PBS as a control. LPS (Sigma-Aldrich) was prepared by dissolving it in PBS to achieve a concentration of 2 mg/mL. Each mouse received an i.p. injection at a dose of 15 mg/kg body weight. Eighteen hours post-injection, mice were euthanized, and tissues were collected for further analysis.

### Monocyte and macrophage depletion in mice

Monocytes and macrophages were depleted by i.v. administration of clodronate liposomes (200 µL per mouse, YEASEN, 40337ES10). Control mice received an equivalent volume of control Liposomes (200 µL per mouse, YEASEN, 40338ES10). Twenty-four hours after liposome injection, mice were administered an i.p. injection of LPS at a dose of 15 mg/kg body weight. The efficiency of monocyte and macrophage depletion was assessed by flow cytometry.

### CLP-induced sepsis model

Following one month of voluntary exercise, mice were randomly assigned to undergo either a moderate-grade CLP procedure or sham operation as a control. To induce moderate-grade sepsis, the distal half of the cecum was ligated with 4–0 silk sutures, followed by a cecal puncture using a 22-gauge needle[62]. The peritoneum, fasciae, abdominal muscles, and skin were closed using interrupted 4–0 silk sutures. Each mouse then received a subcutaneous injection of saline (1 ml/20 g body weight) for fluid resuscitation. Twenty-four hours post-operation, mice were euthanized, and tissues were collected for further analysis.

### Cleavage under targets and tagmentation (CUT&Tag) and CUT&Tag-qPCR

Approximately $1 \times 10^5$ monocytes were collected from mice following one month of voluntary exercise and subsequently 18 h post LPS injection. Monocytes were washed with ice-cold PBS (4 °C). CUT&Tag assays were performed using the NovoNGS CUT&Tag 3.0 High-Sensitivity Kit (for Illumina) (Novoprotein, N259-YH01) according to the manufacturer's instructions. Briefly, monocytes were resuspended in wash buffer and incubated with NovoNGS® ConA Beads for 10 min at room temperature. Samples were then incubated with anti-H3K18la antibody (PTMBIO, PTM-1427RM) for 2 h, followed by incubation with Goat Anti-Rabbit IgG (H&L) secondary antibody (N269) for 1 h, both at room temperature. After washing, samples were incubated with ChiTag® (pAG-Transposome) for 1 h. DNA fragmentation was then performed by adding $MgCl_2$ and incubating for 1 h at 37 °C. The reaction was terminated by adding stop buffer and incubating for 10 min at 55 °C. Genomic DNA was extracted using Tagment DNA Extract Beads (N245). PCR amplification was carried out using indexing primers from the Novoprotein NovoNGS Index Kit for Illumina (N239). The resulting PCR products were purified with NovoNGS DNA Clean Beads (N240). After library preparation as described above, paired-end sequencing was performed on the Illumina NovaSeq 6000 platform.

To further validate the results of CUT&Tag high-throughput sequencing, CUT&Tag-qPCR analysis was performed on Arg1 genes using unmerged libraries. Promoter-specific Arg1 primer pairs (CACCCGGTCGTGGTTCTC and GGTCCGATTGGAACTTGAGTG) were designed. The β-actin (CTCCATCCTGGCCTCGCTGT and GCTGTCACC TTCACCGTTCC) was used as an internal control.

### Echocardiography

Cardiac function in conscious mice was assessed using the Vevo 2100 system (VisualSonics), as previously described[63]. Parasternal standard two-dimensional (2D) and M-mode short-axis view was used

to measure the left ventricular (LV) internal dimensions at diastole (LVIDd) and systole (LVIDs), the LV internal volume at diastole (LVEDV) and systole (LVESV) and the heart rate (HR). The LVEF was calculated as: ((LVEDV − LVESV)/LVEDV) x 100%. the LVFS was determined as ((LVIDd − LVIDs)/LVIDd) x 100%.

## Elisa

IL-1β and TNF-α ELISA kits (Cloud-Clone, USCN Business Co., Ltd) were used to measure cytokine levels in mouse serum according to the manufacturer's instructions.

## Bone marrow-derived macrophages (BMDM)

Macrophages were derived from bone marrow precursors as previously described[15]. Briefly, bone marrow cells ($1.6 \times 10^6$ cells/mL) were cultured in 6 mL of DMEM supplemented with 20% FBS and 30% L929 conditioned medium, which serves as a source of M-CSF, in 10 cm Petri dishes. After 3 days of culture, an additional 3 mL of differentiation medium was added. On the 7th day, macrophages were harvested using ice-cold $Ca^{2+}$ and $Mg^{2+}$-free PBS. The purity of the macrophages was confirmed by flow cytometry (FACS) using the pan-macrophage marker F4/80.

## Peripheral blood mononuclear cells (PBMC)

To obtain peripheral blood mononuclear cells, we first isolated mouse peripheral blood mononuclear cells (PBMC) using a lymphocyte separation solution (YEASON, 40504ES60) through density gradient centrifugation according to the manufacturer's instructions. Subsequently, monocytes were isolated from PBMCs using magnetic cell separation with CD115 magnetic beads (Miltenyi Biotec, Germany) for subsequent experimentation.

## Monocyte reinfusion and tracking

Isolated monocytes were pretreated with 20 mM sodium lactate (Sigma) or PBS as a control for 24 h, and labeled with PKH26 (MKbio) prior to reinfusion. A total of $2 \times 10^6$ PKH26-labeled monocytes were resuspended in 250 µL of PBS and infused into the mice via the tail vein after i.p injection of LPS at a dosage of 15 mg/kg.

## TUNEL staining

Apoptosis was detected by the In Situ Cell Death Detection Kit (Roche), following the manufacturer's instructions. Quantification was performed by Olympus BX41 microscope and CellSense imaging software.

## DHE

Frozen sections of heart tissues were incubated with dihydroethidium (DHE) (YEASEN, Shanghai) to assess oxidative damage, according to the manufacturer's instructions. The mean fluorescent intensity was analyzed using ImageJ Version 10.2.

## Intracellular lactate measurement

Intracellular lactate levels were measured using a lactate assay kit (ab65331, Abcam), following the protocol provided by the manufacturer.

## Histology and immunostainings

Mouse hearts were harvested 18 h post-injection with either LPS or PBS. The tissues were dehydrated in a 30% sucrose gradient for one day, embedded in OCT, and quickly frozen at −80 °C. Sections of 7 µm thickness were prepared and mounted on microscope slides. Immunofluorescence staining was performed on cardiac sections using the following primary antibodies: anti-CD68 (1:250, MCA1957GA, BioRad), anti-Arg1 (1:100, ab133543, Abcam), anti-iNOS (1:200, ab15323, Abcam), anti-Pan Kla (1:100, PTM1401, PTMBio), and anti-cTnT (1:200, 68300, Proteintech). Appropriate secondary antibodies were used: Alexa Fluor 488 Goat anti-Rabbit (Invitrogen, A-11008), Alexa Fluor 555 goat

anti-Rabbit (Invitrogen, A-21428), Alexa Fluor 647 Donkey anti-Rat (Invitrogen, A-21247), Alexa Fluor 488 Donkey anti-Mouse (Invitrogen, A-21202), Alexa Fluor 555 Donkey anti-Rabbit (Invitrogen, A-31572), Goat anti-rabbit DyLight 405 (Invitrogen, SA5-10044). Microscopic analysis was performed using an Olympus BX41 microscope and Cell-Sense imaging software.

## Protein extraction and immunoblot

Cells were lysed in RIPA buffer supplemented with protease and phosphatase inhibitors, then heated at 100 °C for 10 min. For histone extraction, cells were lysed in ice-cold PBS containing 0.5% Triton X-100 and 2 mM PMSF for 10 min. The supernatant was removed by centrifugation at 400 g for 10 min, and the pellet was extracted overnight with 0.2 N hydrochloric acid solution. Protein concentration was determined using a BCA assay kit according to the manufacturer's instructions. Nonspecific binding was blocked with 5% skimmed milk in TBST solution, then incubated overnight at 4 °C with primary antibodies: anti-Pan Kla (1:2000, PTM1401, PTM Bio); anti-Arg1 (1:1000, ab133543, Abcam); anti-iNOS (1:1000, ab15323, Abcam); anti-Histone H3 (1:1000, ab176842, Abcam); anti-β-tubulin (1:5000, 10094, Proteintech); anti-LDHA (1:1000, ab52488, Abcam); anti-H3K18la (1:1000, PTM-1427RM, PTM Bio). Lactyl-Histone Antibody Sampler Kit (PTM-7093, PTM Bio); anti-P300 (1:500, 20695-1, Proteintech); anti-HDAC1(1:1000, 10197-1, Proteintech); anti-HDAC2 (1:5000, 12922-3, Proteintech); anti-HDAC3 (1:1000, 10255-1, Proteintech). Appropriate HRP-conjugated secondary antibodies were incubated for 1 h at room temperature, and signals were visualized using the GE Amersham Imager 600. The uncropped scans of the western blot images are available in the Source data.

## Gene silencing in PBMCs

The silencing of p300 and HDAC1-3 in PBMCs was accomplished through transfection with small interfering RNAs (siRNA) (Gene Pharma Co., Ltd., China). Specifically, a total of $7.5 \times 10^6$ PBMCs were resuspended in 750 µL of Opti-MEM (31985070, Gibco, USA), and then electroporated with 120 pmol of siRNA using the Gene Pulser Xcell™ Electroporation System (165-2661, Bio-Rad, USA) under the following parameters: 250 V, 950 µF, and ∞Ω. As a control, BMDMs were electroporated with scrambled siRNA. The transfection efficiency was validated via WB analysis.

## Human participants

To obtain human peripheral blood monocytes for further RNA sequencing, a total of 19 participants were enrolled in this study, including 10 sepsis patients and 9 non-sepsis controls. Eligible participants met the following criteria: aged between 18 and 85 years. For sepsis cases, fulfillment of the Sepsis-3 definition of sepsis, with concurrent evidence of myocardial injury. Myocardial injury was defined as either elevated high-sensitivity troponin-I (hs-cTnI > 0.04 ng/mL) or echocardiographic evidence of acute left ventricular systolic dysfunction, specifically a left ventricular ejection fraction (LVEF) < 50% with a reduction of ≥10% in LVEF. For non-sepsis controls, volunteers were recruited without a diagnosis of sepsis or evidence of myocardial injury. To obtain human peripheral blood monocytes for histone lactylation, a total of 12 volunteers were enrolled in this study, including 6 volunteers with regular exercise habits, defined as at least 30 min of exercise three times per week, and 6 volunteers with less than one session per week. All participants were recruited from Sir Run Run Shaw Hospital, School of Medicine, Zhejiang University, with informed consent.

## Preparation of nanoparticles

PEG-DSPE (20 mg), DSPE-PEG-NHS (20 mg), and the NOS inhibitor TRIM (26 mg) or Arg1 inhibitor piceatannol 3'-O-glucoside (HY-N2237, MCE) were dissolved in 2 mL tetrahydrofuran and added dropwise to

5 mL PBS solution while stirring for 2 h. After dialyzing the solution in PBS for 24 h (MWCO = 3500), Ly6c antibody (1 mg) was added and stirred for an additional 12 h, followed by dialysis in PBS for another 24 h (MWCO = 30,000). The final volume of the solution was set to 10 mL to create the targeting nano-inhibitor. The drug loading content and antibody grafting content were measured by HPLC and ELISA, and calculated using the following equation:

$$\text{Drug loading content (DLC)} = \frac{M_{TRIM}}{M_{TRIM} + M_{Vehicle}} \qquad (1)$$

$$\text{Antibody grafting content (AGC)} = \frac{M_{Antibody}}{M_{Vehicle}} \qquad (2)$$

The targeting nano-inhibitor was administered by intraperitoneal injection at 300 μL per mouse. For in vitro experiments, cells were treated with the targeting nano-inhibitor at a TRIM concentration of 30 μM.

### Seahorse experiment
Monocytes were isolated from PBMCs as described earlier, and $5 \times 10^5$ cells/mL were seeded into XF96 cell culture microplates (Agilent). Extracellular acidification rates (ECAR) were quantified using the XFe 96 instrument according to the manufacturer's protocol.

### PCR
Total RNA was extracted using the RNA-Quick Purification Kit (Yishan Biotech) and quantified with a Nanodrop 2000 spectrophotometer (Nanodrop Technologies). cDNA was synthesized using the Prime-Script RT Reagent Kit (Takara), and gene expression was analyzed with qPCR using Hieff® qPCR SYBR® Green Master Mix (Yeasen) on the QuantStudio Flex (Thermo). mRNA levels were normalized to 18S rRNA as a housekeeping gene. Primer sequences used were: 18S: 5′-CGGCTACCACATCCAAGGAA-3′ and 5′-GCTGGAATTACCGCGGCT-3′; Arg1: 5′-TTCTCAAAAGGACAGCCTCG-3′ and 5′-CAGACCGTGGGTTCT TCACA-3′; iNOS:5′-GTTCTCAGCCCAACAATACAAGA-3′ and 5′-GTGGA CGGGTCGATGTCAC-3′.

### Bulk RNA-SEQ library preparation and sequencing
RNA purification, reverse transcription, library construction and sequencing were conducted at Shanghai Majorbio Biopharm Technology Co., Ltd. (Shanghai, China). Total RNA from cardiac tissue was extracted using TRIzol reagent and quantified with a NanoDrop spectrophotometer. Libraries were prepared using 1 μg of total RNA and sequenced on the NovaSeq X Plus platform (PE150). Briefly, messenger RNA was isolated using polyA selection with methodbyoligo (dT) beads and then fragmented. Double-stranded cDNA was synthesized using a SuperScript double-stranded cDNA synthesis kit (Invitrogen, CA) using random hexamer primers. The synthesized cDNA was subjected to end-repair, phosphorylation, and adapter ligation, according to the library construction protocol. Libraries were size selected for 300 bp cDNA fragments using 2% Low Range Ultra Agarose, followed by PCR amplified with Phusion DNA polymerase (NEB) for 15 cycles. The sequencing library was quantified with Qubit 4.0 and sequenced using the NovaSeq Reagent Kit on NovaSeq X Plus platform (PE150).

### Analysis of bulk RNA-seq data
Initial reads QC metrics (base quality distribution) were assessed using FASTQC. NGS QC toolkits were used to trim adapters and low-quality reads. The clean reads were mapped to the human (hg19) and mouse (mm10) genomes using HISAT2 version 2.2.1 with default settings. BAM files containing uniquely mapped reads were used as inputs for the Stringtie, and transcripts per million reads values were calculated to quantify gene expression levels. Differential gene expression analysis was performed using the DESeq2 package in R, and DEGs were

identified with the criteria of $|\log 2FC > 1|$ and adj.pvalue < 0.05. Immune infiltration analysis was performed using ImmuCellAI (http://bioinfo.life.hust.edu.cn/ImmuCellAI-mouse/#!/).

### FACS sorting of CD45⁺ immune cells from mouse hearts
Mice were anesthetized using isoflurane inhalation, and the hearts were perfused with 20 mL of cold PBS to remove peripheral blood. Atria and valves were removed, and the ventricles were minced into around 1 mm cubes. Heart tissues were digested in a solution containing 2 mg/mL collagenase IV, 1 mg/mL dispase II, and 60 U/mL DNase at 37 °C for 40 min with shaking. After dissociation, the cell suspensions were filtered through a 40 μm strainer, followed by red blood cell lysis (Sigma) and dead cell removal (Miltenyi Biotec, 130-090-101). As previously described[64], single-cell suspensions were incubated with CD45 MicroBeads (Miltenyi Biotec, 130-052-301) at 4 °C for 10 min and isolated using a MACS Separator (Miltenyi Biotec). The cell layer between the liquid surfaces was collected and washed with 10 mL of PBS for further experiments.

### Single-cell library preparation and sequencing
Isolated Cd45⁺ cells were washed, resuspended in cold PBS, and processed on the Chromium Single Cell Gene Expression platform (10X Genomics). The Single Cell 3′ Reagent Kit v2 facilitated droplet formation with single cells and barcoded beads (10X Genomics). Following cell lysis, reverse transcription was performed using a thermal cycler (ProFlex PCR). cDNA was purified utilizing Dynabeads (10X Genomics) and amplified for 10 or 14 cycles within the thermal cycler (ProFlex PCR). The cDNA was fragmented, ligated with adapters and sample index, and selected using SPRI beads (Beckman). Libraries were sequenced on the NovaSeq 6000 Illumina sequencing platform, and raw reads were aligned to the mm10 reference genome using Cell-Ranger version 6.1.1 (10X Genomics).

### Quality control, dimension reduction, clustering, and differentially expressed gene expression analysis
CellRanger outputs were loaded into Seurat (v.4.4.0) for unsupervised clustering[65]. Genes expressed in fewer than 3 cells, cells with fewer than 200 genes, unique molecular identifier counts less than 300, and mitochondrial DNA (mtDNA) expression exceeding 30% were excluded. Doublets were removed using DoubletFinder (version 2.0.2)[66]. The "LogNormalize" method in Seurat was used for normalization. The FindVariableGenes function was used to select the top 3000 variable genes for downstream data integration, features and anchors, based on the functions of "FindIntegrationAnchors" and "IntegrateData". The normalized expression was scaled through the function of "ScaleData" to remove unwanted sources of variation. Principal component analysis (PCA) was performed, and the top 30 components (PCs) were used for UMAP dimensionality reduction. Clusters were identified using "FindClusters" and annotated based on cluster-specific genes detected by "FindAllMarkers" function (min.pct = 0.1, logfc.threshold = 0.25) and classical immune cell marker genes. CD45⁻ cells were filtered out to focus on immune cells.

### Pseudotime analysis
Monocle v.2.4.0 was utilized to explore developmental trajectories between macrophage and monocyte subsets[67]. Dimensionality reduction was executed using the "reduceDimension" function with the "DDRTree" method. The "DifferentialGeneTest" function was employed to infer DEGs from each cluster, and the trajectories were visualized using the "plot_cell_trajectory" function. The "root_state" (the starting point) of the trajectory was defined by the higher expression of monocytic genes. After constructing the cell trajectories, DEGs along the pseudotime were identified using "DifferentialGeneTest". Changes in branch-dependent genes with the lowest q values were visualized using plot_pseudotime_heatmap. The

dynamic trends in expression levels of selected significant genes through pseudotime were illustrated using individual graphs generated by plot_genes_in_pseudotime.

### Cell−cell communications analysis

CellChat (version 1.1.3) was employed to elucidate intercellular communication through ligand-receptor interaction networks[68]. A curated database of ligand-receptor interactions, supported by gene expression data, was integrated. Advanced computational techniques, including social network analysis, manifold learning, and pattern recognition, were utilized to dissect these interactions. Probability models of communication between distinct cell types were constructed, and representative ligand-receptor pairs were selected based on a statistical significance threshold of $P < 0.01$.

### Gene ontology enrichment and gene set score analysis

Gene ontology (GO) enrichment analysis was performed using the clusterProfiler R package. Enrichment results were selected based on a statistical threshold (qvalueCutoff = 0.05), focusing on biological process (BP). Publicly available gene sets were obtained from Molecular Signatures Database (MSigDB)[69]. The "AddModuleScore" function in Seurat was used to score gene sets for each input cell.

### Metabolic flux balance analysis

For the flux balance analysis using the Compass approach, the expression matrix and associated metadata were extracted from the Seurat object[70]. Data micropooling was implemented with the Vision package (version 2.1.0), generating pooled cells cellsPerPartition set to 10, significantly reducing computational time. The micropooled expression matrix and metadata were used for Compass analysis, conducted with the default metabolic model settings for Mus_musculus and the RECON2_mat framework. Subsequent data postprocessing was performed in Python, following the Compass tutorial protocols.

### Statistical analysis

Statistical analyses were conducted utilizing GraphPad Prism v9.0. Normally distributed data were compared using the unpaired two-tailed Student's t-test, while non-normally distributed data were analyzed with the Mann–Whitney test. Results are presented as the mean ± standard error of the mean (s.e.m.) for parametric data or median with interquartile range for non-parametric data.

### Reporting summary

Further information on research design is available in the Nature Portfolio Reporting Summary linked to this article.

### Ethical approval

All experimental animal procedures were approved by the Animal Care and Use Committee of the School of Medicine, Zhejiang University (Approval number: ZJU20230056). All experimental protocols using patient blood samples were approved by the Ethics Committee of Sir Run Run Shaw Hospital, School of Medicine, Zhejiang University (Approval numbers: 2024-0409, 2025-0443).

## Data availability

The sequencing data generated in this study were deposited at the National Center for Biotechnology Information's Sequence Read Archive (SRA) and National Genomics Data Center (NGDC) under accession numbers PRJNA1172003, PRJNA1171952, PRJNA1300856, and PRJCA051457. Further information on research design is available in the Nature Portfolio Reporting Summary linked to this article. Source data are provided with this paper.

## Code availability

This study did not involve the development of custom code. All data analyses were conducted using existing, publicly available tools and packages, which are cited and described in detail in the methods.

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

## Acknowledgements

This work was supported National Natural Science Foundation of China: 82570417 (M.S.), 82200271 (M.S.), 82311530688 (M.S.), Key Research and Development Program of Zhejiang Province 2025C02163 (M.S.), Natural Science Funds of Zhejiang Province LQ22H020010 (M.S.), Leading Innovation and Entrepreneurship Team of Zhejiang Province 2023R01005 (M.S.), Natural Science Funds of Shandong Province ZR2024QH544 (S.S.), Shandong Province Medical and Health Science and Technology Project 202303011354 (S.S.). Schematics were generated with Biorender. We thank Xiaoli Hong and Chao Bi from the Core Facilities, Zhejiang University School of Medicine, for their technical support. We thank OE Biotech for providing single-cell RNA-Seq.

## Author contributions

S.S. analyzed and interpreted the data, and drafted the manuscript. C.L. performed the majority of experiments. C.H. prepared single-cell suspensions for scRNA seq experiments. X.R. and T.Z. performed the CLP experiment. J.Z. conducted the CUT&Tag experiment. Y.T. and Q.Z. performed histology and mouse treatment. J.L. performed Seahorse measurements. Z.S. recruited volunteers and collected blood samples. W.C. assisted in bioinformatic analysis. R.W. and N.R. performed in vitro assays. X.W. designed the CUT&Tag experiment. B.M. designed and synthesized the nanoparticles used in the study. J.Q. contributed to scRNA seq data analysis and interpretation and critically revised the manuscript. G.F. critically revised the manuscript. M.S. supervised the project, performed the experimental design and data interpretation, provided scientific direction, and wrote the manuscript.

## Competing interests

The authors declare no competing interests.
