## [Transparent Peer Review file · Nature Communications]

Exercise-Induced Histone Lactylation in Monocyte-Derived Macrophages Restores Cardiac Immune Homeostasis and Function in Sepsis-Induced Cardiomyopathy

Corresponding Author: Professor Min Shang

Version 0:

Reviewer comments:

Reviewer #1

(Remarks to the Author)

In this manuscript, the authors emphasize the role of voluntary running in preserving cardiac function and immune homeostasis in sepsis. Through a series of meticulously designed experiments, they elucidated the protective mechanisms of exercise against sepsis-induced cardiomyopathy (SICM). The study identified a distinct subpopulation of monocyte-derived cardiac macrophages, designated as iNOS⁺ Arg1⁺, which exhibited both inflammatory and wound-healing gene expression profiles. The results showed that inhibition of the pro-inflammatory iNOS in monocyte-derived macrophages impaired the exercise-induced preservation of cardiac function. Furthermore, the authors demonstrated that exercise enhanced glycolysis in monocytes, leading to increased lactate production. This, in turn, promoted histone lactylation, which accelerated the transition of cardiac macrophages from a pro-inflammatory to a pro-reparative state, thereby restoring cardiac immune homeostasis and maintaining cardiac function in SICM. The reviewers found this discovery to be intriguing and one of the major highlights of the study. However, the discovery, identification, and elucidation of the specific mechanisms underlying the lactylation modification remain preliminary. Therefore, despite the interesting nature of this finding, the authors are encouraged to provide more robust experimental evidence to support their conclusions. In conclusion, the work holds considerable potential, but its acceptance is contingent upon a more thorough elucidation of the lactylation mechanism and its functional implications in SICM.

Major criticisms:

1. The authors cite recent research findings to demonstrate that "glycolysis-derived lactate can modulate histone lactylation, thereby promoting wound-healing gene expression," which provides a critical link between exercise and the anti-inflammatory and reparative functions of cells. This is because the authors have previously shown that exercise enhances glycolysis and lactate production. Based on this evidence, exercise would further influence histone lactylation and the expression of related genes. However, to validate this hypothesis, the authors only relied on methods such as Western Blotting and immunofluorescence, which are insufficient to fully substantiate this claim. More advanced methodologies, such as mass spectrometry, could be employed to precisely characterize the relationship between exercise and histone lactylation. Additionally, in the study of histone lactylation, the authors used Pan-K1a antibodies but failed to specify which histone underwent lactylation. More importantly, the exact lactylation sites on the histones remain unidentified. The authors should further explore these specific mechanisms, including the identification of lactylated histone subtypes and the precise lactylation sites, which would enhance the scientific rigor of the study and make the findings more compelling and engaging for readers.

2. To further investigate the role of exercise-induced histone lactylation in monocyte derived macrophages in sepsis-induced cardiomyopathy (SICM), the authors generated mice lacking LDHA lactate dehydrogenase A (LDHA) specifically in myeloid cells. The authors suggest that LDHA is a key enzyme in anaerobic glycolysis that converts pyruvate to lactate. In SICM patient monocytes, LDHA expression is significantly increased, and exercise-enhanced glycolysis in monocytes increases lactate production, thereby promoting histone lactylation. Knockdown of the LDHA gene can block this critical step in lactate production, clarifying the role of the LDHA-mediated glycolysis-lactate production-histone lactylation pathway in the disease. In fact, in the article published in 2022 (Sci Adv. 2022, 8(3): eabi6696.), it was clearly pointed out that HDAC1-3 are histone "erasers". In the article published in 2024 (Mol Cancer. 2024, 23(1):90.), P300 was identified as a potential "writer" of histone lactylation. In this paper, when verifying the role of exercise-induced histone lactylation in monocyte-derived macrophages in SICM, the authors should adopt a more direct approach by interfering with the "writers" and "erasers" of histone lactylation to further investigate the role of exercise-induced histone lactylation in monocyte-derived

macrophages in SICM.

Minor criticisms:

- 1、 There are many elementary mistakes in this manuscript, which might leave a bad impression on this work. The authors should carefully check this manuscript and revised it. For example:
 - a) The full title cited in the letter to the editor regarding SICM should be "Sepsis-Induced Cardiomyopathy" instead of "Sepsis-Induced Cardiomyopath." I hope the entire text is reviewed to avoid such errors;
 - b) Page 2-line 14: the abbreviation SICM appears for the first time in the abstract, but its full form is not provided.
 - c) Figure 4R: Regarding the "ARG1+CD68+ Positive Area (%)", please note that "ARG1+" is different from "Arg1+" in other figures. Please check and standardize this.
 - d) Page 12-line 22: In the main text, the authors pointed out that the global histone lactylation induced by exercise in both circulating monocytes and cardiac macrophages in SICM can be verified in Figures 5F to 5L. However, we could not find Figures 5J to 5L in Figure 5.
- 2、 In the study of sepsis-induced cardiomyopathy (SICM), the authors used the lipopolysaccharide (LPS)-induced modeling method. However, from the perspective of simulating clinical conditions, the cecal ligation and puncture (CLP) model may be a more advantageous modeling method. Moreover, a study published in 2023 (Nat Metab. 2023, 5(1):129-146.) also employed the CLP method to induce SICM. Therefore, it is hoped that the authors can provide a more robust justification for using the LPS-induced SICM disease model.
- 3、 To validate the mechanism that exercise boosts glycolysis and subsequent histone acetylation in monocyte-derived macrophages during sepsis-induced cardiomyopathy (SICM), this study isolated peripheral blood monocytes from both SICM patients and healthy donors using CD14+ beads and performed RNA-seq analysis. However, the authors didn't specify which healthy samples were used in the methods part. The study would benefit from a table with patient information and clarification on SICM diagnosis criteria
- 4、 In fact, animal experiments should include the animal ethics approval number in scientific research. However, we did not find the relevant ethics number in the manuscript. Please check and add it.
- 5、 The discussion provides a good overview of the study's findings in relation to the broader field. However, it could be improved by discussing potential limitations of the study more explicitly and how these might affect the interpretation of the results.

Reviewer #2

(Remarks to the Author)

This is an exciting study, exploring the effects of exercise on monocytes and macrophages, and eventually their effects on sepsis-induced cardiomyopathy. Using different elegant and complementary tools, the authors come up with a mechanism including a metabolic shift in monocytes towards glycolysis that results in increased histone lactylation, and the appearance of iNOS/Arg1+ macrophages.

While some of the questions I have result from pure interest, other questions relate to experimental issues and the lack of crucial controls and experiments to support all claims being made. Another general aspect to mention is the lack of clear human relevance and validation. While the authors did do some analyses on human cells, only LDHA is validated and an open question remains to what extent iNOS/Arg1 etc are relevant for the human situation (based on the current literature they are not), and if exercise indeed also elicits a metabolic shift and K1a in monocyte in humans.

Specific questions and concerns:

- The effects on monocytes are likely acute and fast since these cells are short-lived? How can this be aligned with the observed effects on macrophages within the heart? The authors now perform a 3-month study, but in principle if exercise-induced lactate indeed mediates the effects via monocytes, one session of exercise should elicit similar effects. Such experiment with shorter time points would help to support the current working scheme and hypothesis. Another aspect that should be explored is if exercise induces shifts in tissue resident macrophages as they can also contribute to the observed effects. Importantly, the effects of exercise should also explored before LPS stimulation.
- Related to the latter point, key controls (PBS + Exe) are missing in quite some of the assays (e.g. bulk and scRNAseq) and as such once cannot dissect which effects are truly induced by exercise perse. These data are crucial to answer the key question of how monocyte and macrophage profiles are modulated by exercise (before induction of sepsis). It looks like the sc analyses did not reveal any significant differences. Were ECAR levels also increased in exercised mice without sepsis? If that is not the case, I also don't see the point of lactate-education and monocyte transfer.
- While LPS is often used to mimic sepsis in vivo, it does not truly induce sepsis. Therefore, key findings should be validated in a proper sepsis model to support the claims being made (e.g. clp and/or cecal slurry injection)
- Based on a drop in IL-6 and IL-10, the authors claim an attenuated pro-inflammatory response. Yet, IL-10, and to some extent also IL-6, are anti-inflammatory cytokines. As such, key inflammatory mediators like IL-1b, TNF etc should be measured to support this claim. Strikingly, IL-6 is also a myokine induced by muscle and it is therefore surprising to see it being reduced in the Exe group. Maybe timing wasn't ideal? The authors should at least discuss this aspect.
- While the authors do show the importance of iNOS in mediating the observed effects (using an elegant approach with inhibitor/nanoparticle), such proof for Arg1 is missing and should be provided to support all claims being made. The overall working hypothesis is that lactate-induced monocytes from the circulation end up in the heart and become Arg1-expressing cells that do the job? Is there actual proof of this? There are very good anti-Arg1 antibodies available for flow cytometry so it would be relatively easy to see if transferred monocytes become Arg1+ cells. Actual proof should come from the use of KO models to support this hypothesis. Now Arg1 is rather a marker, and not yet a mediator.
- Bulk sequencing appears a bit redundant since the authors also performed scRNAseq. The rationale of using both should be better explained, and also similarities and differences in the results should be highlighted. With regards to differences being observed in the indicated genes in bulk, analyses, it would be important to use the sc data to dissect which cell types are responsible for the observed effects in bulk analyses

- The authors now did a transfer of lactate-educated monocytes. Transfer monocytes from trained mice to naive ones before the induction of sepsis should be performed to demonstrate that it are indeed the training of monocytes mediating the effects. This is important as also resident macrophages will be influenced by the exercise and likely also contribute to the observed phenotype.
- The authors mention that resident macrophage markers like Apoe, C1qa, Mrc1 were upregulated during differentiation from Ly6c Mo to MCHII hi macrophages. I would say monocytes do not differentiate into tissue-resident macrophages so (i) the markers are not good markers, and/or (ii) the pseudotime analysis makes a trajectory that is not actually there.
- Ldha show decreased K1a but also other effects. To what extent does the lactylation play a role? Further (albeit also indirect) proof could be provided by using inhibitors that block histone lactylation.
- Overall, the human relevance of the current findings are very limited and key aspects should be validated in the human setting. For example; is lactylation also increased in human monocytes by exercise?

Version 1:

Reviewer comments:

Reviewer #1

(Remarks to the Author)

In the revised manuscript, the authors have made commendable efforts to improve their data, (especially for in-depth analysis of the characterization of exercise-induced histone lactylation). By optimizing the experimental design and supplementing key data, the authors have effectively clarified the detailed mechanisms underlying the core findings of the study. In particular, the logical chain of the core regulatory axis—"Exercise-induced lactate triggered H3K181a, effectively pre-trains monocyte derived macrophages to preadapt to sepsis before cardiac recruitment in SICM"—has become more comprehensive. Most of my previous concerns have been addressed, but a few issues remain that I believe still need attention:

1、 Although this study has included sepsis patients and analyzed the metabolic characteristics of their monocytes, it has not detected the expression of Pan-K1a and H3K181a in monocytes from sepsis patients (especially those with concurrent SICM). Nor has it established associations between these lactylation indicators and patients' cardiac function prognosis (e.g., ejection fraction [EF] value, mortality rate). This leads to a lack of key correlative evidence for the conclusion that "lactylation regulation improves SICM" in the pathological context of human sepsis, thereby weakening the persuasiveness of its clinical translation. The author needs to provide some explanations in discussion.

2、 There are many elementary mistakes in this manuscript, which might leave a bad impression on this work. The authors should carefully check this manuscript and revised it. For example:

- a) There is inconsistency in the expression of the core term "monocyte derived macrophages" throughout the manuscript—hyphens are omitted in some paragraphs (where it is written as "monocyte-derived macrophages"). It is recommended to standardize the term uniformly to comply with academic writing conventions and avoid ambiguity.
- b) There is a term confusion between "cardiovascular diseases (CVD)" in the abstract and "cardiovascular diseases (CAD)" in the introduction. CVD (Cardiovascular Diseases) is a general term for cardiovascular disorders (and SICM, i.e., sepsis-induced cardiomyopathy, falls within this category), while CAD specifically refers to Coronary Artery Disease. These two concepts are distinct, which may easily cause readers to misunderstand the scope of the study. It is necessary to uniformly revise the term to "cardiovascular diseases (CVD)".

Reviewer #2

(Remarks to the Author)

The authors addressed to most important concerns and I now consider this article suitable for publication.

Version 2:

Reviewer comments:

Reviewer #1

(Remarks to the Author)

The authors have added adequate descriptions to answer my queries. I think this paper can now go to the next step for publication.

Response to Reviewer 1:

In this manuscript, the authors emphasize the role of voluntary running in preserving cardiac function and immune homeostasis in sepsis. Through a series of meticulously designed experiments, they elucidated the protective mechanisms of exercise against sepsis-induced cardiomyopathy (SICM). The study identified a distinct subpopulation of monocyte-derived cardiac macrophages, designated as iNOS⁺ Arg1⁺, which exhibited both inflammatory and wound-healing gene expression profiles. The results showed that inhibition of the pro-inflammatory iNOS in monocyte-derived macrophages impaired the exercise-induced preservation of cardiac function. Furthermore, the authors demonstrated that exercise enhanced glycolysis in monocytes, leading to increased lactate production. This, in turn, promoted histone lactylation, which accelerated the transition of cardiac macrophages from a pro-inflammatory to a pro-reparative state, thereby restoring cardiac immune homeostasis and maintaining cardiac function in SICM. The reviewers found this discovery to be intriguing and one of the major highlights of the study. However, the discovery, identification, and elucidation of the specific mechanisms underlying the lactylation modification remain preliminary. Therefore, despite the interesting nature of this finding, the authors are encouraged to provide more robust experimental evidence to support their conclusions. In conclusion, the work holds considerable potential, but its acceptance is contingent upon a more thorough elucidation of the lactylation mechanism and its functional implications in SICM.

Response: We sincerely thank the reviewer for his/her thoughtful and detailed evaluation of our study. We are particularly grateful for the precise and logical summary of our work, as well as for recognizing the intriguing nature and significant potential of our findings. We are delighted that the reviewer appreciates the novelty of our study, particularly the identification of the iNOS⁺ Arg1⁺ macrophage subpopulation and the role of exercise-induced histone lactylation in preserving cardiac function and immune homeostasis in SICM. We greatly respect and value his/her constructive feedback and in response to the reviewer's comments, we have conducted additional experiments and analyses to provide more robust experimental evidence, further strengthening the mechanistic insights and functional implications of histone lactylation in SICM. Once again, we sincerely thank the reviewer for his/her insightful comments, which have greatly contributed to improving the quality of our manuscript. We sincerely hope this revised version of manuscript will meet the reviewer's expectations and provide valuable insights for future readers.

Major criticisms:

1. The authors cite recent research findings to demonstrate that "glycolysis-derived lactate can modulate histone lactylation, thereby promoting wound-

healing gene expression," which provides a critical link between exercise and the anti-inflammatory and reparative functions of cells. This is because the authors have previously shown that exercise enhances glycolysis and lactate production. Based on this evidence, exercise would further influence histone lactylation and the expression of related genes. However, to validate this hypothesis, the authors only relied on methods such as Western Blotting and immunofluorescence, which are insufficient to fully substantiate this claim. More advanced methodologies, such as mass spectrometry, could be employed to precisely characterize the relationship between exercise and histone lactylation. Additionally, in the study of histone lactylation, the authors used Pan-Kla antibodies but failed to specify which histone underwent lactylation. More importantly, the exact lactylation sites on the histones remain unidentified. The authors should further explore these specific mechanisms, including the identification of lactylated histone subtypes and the precise lactylation sites, which would enhance the scientific rigor of the study and make the findings more compelling and engaging for readers.

Response: We thank the reviewer for the insightful comment and acknowledge that a more in-depth analysis of the characterization of exercise-induced histone lactylation would be more convincing. To address these concerns, we have used additional approaches.

i. To identify the specific histone lactylation sites induced by exercise in monocytes, also considering that so far the identified histone lactylation sites are mainly on histone H3 and H4^{1,2}, as previously described³. We analyzed several well-characterized lactylation sites, including H3K18la, H3K9la, H3K14la, H4K5la, H4K8la, H4K12la and H4K16la, in peripheral blood monocytes from mice. Among these, H3K18la levels were significantly elevated in exercise mice compared to sedentary controls treated with PBS as a control, and were even more pronounced in LPS-challenged mice (Figure 7A-C). In contrast, other lactylation sites, including H3K9la, H3K14la, H4K5la, H4K8la, H4K12la, and H4K16la, did not show significant changes in response to exercise under either LPS challenge or PBS control conditions (Figure 7A, D-I). Notably, the pattern of H3K18la closely mirrored the Pan-Kla pattern, indicating that exercise specifically induces H3K18la.

ii. To investigate whether regular exercise similarly promotes H3K18la in human monocytes, we recruited volunteers with sedentary and active lifestyles (Figure 7J, Table S3). Given that lactate levels influence histone lactylation, as reported in previous studies^{3,4}, we treated monocytes from sedentary volunteers with exogenous lactate and observed increased levels of Pan-Kla and H3K18la (Figure 7K-M). Consistently, monocytes from active volunteers exhibited significantly higher levels of Pan-Kla and H3K18la compared to those from sedentary individuals (Figure 7N-P). These findings confirm that exercise-induced lactate accumulation promotes H3K18la in

human monocytes.

iii. Bearing the reviewer's suggestion to employ more advanced methodologies, we utilized genome-wide Cleavage Under Targets and Tagmentation (CUT&Tag) analysis with an anti-H3K18la antibody, followed by high-throughput DNA sequencing, to investigate the regulatory role of H3K18la in *Arg1* expression. The results revealed significant enrichment of H3K18la in the promoter regions of *Arg1* (Figure 8P). Furthermore, CUT&Tag-qPCR analysis demonstrated increased H3K18la enrichment in the promoter regions of *Arg1* following LPS stimulation compared to PBS injection, with even greater enrichment observed in the context of regular exercise (Figure 8Q).

In summary, these findings highlight that exercise-induced H3K18 histone lactylation in monocytes from both mice and humans subsequently promotes *Arg1* expression. This establishes a critical link between exercise and the anti-inflammatory and reparative functions of macrophages in SICM.

2. To further investigate the role of exercise-induced histone lactylation in monocyte derived macrophages in sepsis-induced cardiomyopathy (SICM), the authors generated mice lacking LDHA lactate dehydrogenase A (LDHA) specifically in myeloid cells. The authors suggest that LDHA is a key enzyme in anaerobic glycolysis that converts pyruvate to lactate. In SICM patient monocytes, LDHA expression is significantly increased, and exercise-enhanced glycolysis in monocytes increases lactate production, thereby promoting histone lactylation. Knockdown of the LDHA gene can block this critical step in lactate production, clarifying the role of the LDHA-mediated glycolysis-lactate production-histone lactylation pathway in the disease. In fact, in the article published in 2022 (Sci Adv. 2022, 8(3): eabi6696.), it was clearly pointed out that HDAC1-3 are histone "erasers". In the article published in 2024 (Mol Cancer. 2024, 23(1):90.), P300 was identified as a potential "writer" of histone lactylation. In this paper, when verifying the role of exercise-induced histone lactylation in monocyte-derived macrophages in SICM, the authors should adopt a more direct approach by interfering with the "writers" and "erasers" of histone lactylation to further investigate the role of exercise-induced histone lactylation in monocyte-derived macrophages in SICM.

Response: We thank the reviewer for the valuable suggestion to investigate the roles of histone lactylation "writers" and "erasers" and the role of exercise-induced histone lactylation in protecting against SICM. To address these points, we have conducted the following experiments:

i. We observed significant increases in global Pan-K1a and H3K18la levels in

monocytes isolated from exercised mice injected with LPS or PBS as a control, human volunteers with or without regular exercise habits, and lactate-educated monocytes from mice and human volunteers *in vitro* (Figure 7A-P, Figure S9E, F). These findings highlight the robust induction of histone lactylation at H3K18 by exercise in both mice and humans.

ii. As stated by the reviewer, p300 has been reported as a potential “writer” of histone lactylation⁵. To validate its role in exercise-induced lactylation, we silenced p300 in mouse peripheral blood monocytes using siRNA and incubated them with exogenous lactate to induce lactylation⁶. Consistently, lactate-induced histone lactylation at H3K18la was abrogated when p300 was silenced (Figure 8D-F). These data confirm that p300 functions as the histone lactylation “writer” for H3K18la in monocytes.

iii. To further explore the role of p300 in exercise-induced histone lactylation and its protective effects against SICM, we evaluated the *in vivo* impact of p300-inhibited monocytes in SICM. Primary monocytes were pretreated with C646, the selective inhibitor of p300⁷, before adoptive transfer into SICM mice. Mice receiving p300-inhibited monocytes exhibited an abrogation of the protective effects of exercise, including diminished improvements in body temperature, weight loss, cardiac function, and myocardial apoptosis (Figure S13A-J). These findings indicate that p300 acts as the histone lactylation “writer” of exercise-induced H3K18la and plays a critical role in protecting cardiac functions in SICM.

iv. To identify the “eraser” of histone lactylation, we focused on HDAC1-3, which have been reported as potential “erasers” of histone lactylation⁸. Using siRNA, we silenced HDAC1, HDAC2, and HDAC3 in primary monocytes and evaluated lactate-induced H3K18la levels. Knockdown of HDAC2 resulted in a marked increase in H3K18la levels (Figure 8G-I), whereas silencing HDAC1 or HDAC3 did not significantly increase H3K18la levels (Figure S11A-F). These results identify HDAC2 as the principal “eraser” of H3K18la in monocytes.

Together, these additional experiments demonstrate that p300 is the “writer” and HDAC2 is the “eraser” of exercise-induced histone lactylation in monocyte-derived macrophages, regulating its protective effects in SICM.

Minor criticisms:

1. There are many elementary mistakes in this manuscript, which might leave a bad impression on this work. The authors should carefully check this manuscript and revised it. For example:

a) The full title cited in the letter to the editor regarding SICM should be "Sepsis-

Induced Cardiomyopathy" instead of "Sepsis-Induced Cardiomyopath." I hope the entire text is reviewed to avoid such errors;

Response: We apologize for this oversight. This error has been corrected in the revised version of the letter to the editor. Additionally, we have thoroughly reviewed the entire manuscript to identify and correct any remaining typographical errors, ensuring accuracy and clarity.

b) Page 2-line 14: the abbreviation SICM appears for the first time in the abstract, but its full form is not provided.

Response: Thank you for noting this oversight. In the revised manuscript, we have added the full term "sepsis-induced cardiomyopathy (SICM)" where the abbreviation first appears in the abstract. Furthermore, we have reviewed all other abbreviations to ensure consistent definition throughout the text.

c) Figure 4R: Regarding the "ARG1+CD68+ Positive Area (%)", please note that "ARG1+" is different from "Arg1+" in other figures. Please check and standardize this.

Response: We appreciate the reviewer's attention to the inconsistency in gene symbol formatting. In the revised manuscript, we have corrected the label in Figure 4R to "Arg1+CD68+ Positive Area (%)" and conducted a comprehensive review of all figures, tables, and text to ensure uniform use of "Arg1+" throughout the manuscript.

d) Page 12-line 22: In the main text, the authors pointed out that the global histone lactylation induced by exercise in both circulating monocytes and cardiac macrophages in SICM can be verified in Figures 5F to 5L. However, we could not find Figures 5J to 5L in Figure 5.

Response: We apologize for the confusion caused by the incorrect reference to Figures 5J-5L in the main text. The correct reference should be Figures 5F-I instead of Figures 5F-L. This error has been addressed in this revised version of manuscript (Page 14, Line 14).

2. In the study of sepsis-induced cardiomyopathy (SICM), the authors used the lipopolysaccharide (LPS)-induced modeling method. However, from the perspective of simulating clinical conditions, the cecal ligation and puncture (CLP) model may be a more advantageous modeling method. Moreover, a study published in 2023 (Nat Metab. 2023, 5(1):129-146.) also employed the CLP method to induce SICM. Therefore, it is hoped that the authors can provide a more robust justification for using the LPS-induced SICM disease model.

Response: We thank the reviewer for this important suggestion. Following this reviewer's comments, we have provided a more detailed explanation for

focusing on the LPS-induced SICM model in the introduction of this revised manuscript. Specifically, lipopolysaccharide (LPS), a potent endotoxin derived from Gram-negative bacteria, plays a pivotal role in the pathogenesis of sepsis and its associated organ failure⁹⁻¹¹. This makes the LPS-induced model highly relevant for studying the mechanisms underlying SICM. To further enhance the translational relevance of our findings, we have now employed the cecal ligation and puncture (CLP) model, which is more clinically representative of polymicrobial sepsis¹². Importantly, consistent with our observations in the LPS-induced model, exercise confers significant protection against cardiac dysfunction in the CLP model (Figure S12A-K). These results reinforce the conclusions drawn from our findings across different sepsis models and further highlight the protective effects of exercise in SICM.

3. To validate the mechanism that exercise boosts glycolysis and subsequent histone acetylation in monocyte-derived macrophages during sepsis-induced cardiomyopathy (SICM), this study isolated peripheral blood monocytes from both SICM patients and healthy donors using CD14+ beads and performed RNA-seq analysis. However, the authors didn't specify which healthy samples were used in the methods part. The study would benefit from a table with patient information and clarification on SICM diagnosis criteria

Response: We thank the reviewer for highlighting the need to clarify our control cohort and the diagnostic criteria for SICM. In response, we have revised the methods section and added a new **Table S2** to provide detailed information about the study participants. Specifically, Table S2 outlines the age and sex of both SICM patients and healthy donors, as well as ejection fraction values and the sequential organ failure assessment (SOFA) score for SICM patients. Furthermore, we have clarified the diagnostic criteria for SICM in the methods section. Eligible participants met the following criteria: aged between 18 and 85 years. For sepsis cases, fulfillment of the Sepsis-3 definition of sepsis, with concurrent evidence of myocardial injury. Myocardial injury was defined as either elevated high-sensitivity troponin-I (hs-cTnI > 0.04 ng/mL) or echocardiographic evidence of acute left ventricular systolic dysfunction, specifically a left ventricular ejection fraction (LVEF) < 50% with a reduction of ≥ 10% in LVEF. For non-sepsis controls, volunteers were recruited without a diagnosis of sepsis or evidence of myocardial injury.

4. In fact, animal experiments should include the animal ethics approval number in scientific research. However, we did not find the relevant ethics number in the manuscript. Please check and add it.

Response: We thank the reviewers for their professional comments. In response, we have added the animal ethics approval number (ZJU20230056) to the updated manuscript. We apologize for the omission and appreciate the opportunity to correct it.

5. The discussion provides a good overview of the study's findings in relation to the broader field. However, it could be improved by discussing potential limitations of the study more explicitly and how these might affect the interpretation of the results.

Response: Thanks to the reviewer's suggestions, we realized that it is necessary to have an in-depth discussion on the limitations of this study. Based on the reviewers' comments, we supplemented the discussion on the limitations of this study in the following aspects:

i. We evaluated the role of exercise and monocyte-derived macrophages in cardiac function only 18 hours after LPS injection. Future studies incorporating additional time points following LPS stimulation could provide valuable insights into their role in long-term cardiac remodeling.

ii. While we demonstrated that monocytes from active volunteers exhibited significantly higher levels of both Pan-K1a and H3K181a compared to those from sedentary individuals, consistent with conclusions reached in the mouse study, technical limitations prevented us from fully exploring the role of histone lactylation in SICM among sepsis patients. Its therapeutic potential warrants further investigation through clinical trials.

Response to Reviewer 2:

This is an exciting study, exploring the effects of exercise on monocytes and macrophages, and eventually their effects on sepsis-induced cardiomyopathy. Using different elegant and complementary tools, the authors come up with a mechanism including a metabolic shift in monocytes towards glycolysis that results in increased histone lactylation, and the appearance of iNOS/Arg1+ macrophages.

While some of the questions I have result from pure interest, other questions relate to experimental issues and the lack of crucial controls and experiments to support all claims being made. Another general aspect to mention is the lack of clear human relevance and validation. While the authors did do some analyses on human cells, only LDHA is validated and an open question remains to what extent iNOS/Arg1 etc are relevant for the human situation (based on the current literature they are not), and if exercise indeed also elicits a metabolic shift and K1a in monocyte in humans.

Response: We sincerely thank the reviewer for the thoughtful and detailed evaluation of our study, as well as for the positive remarks regarding the novelty and approach of our work. We deeply appreciate the recognition of our study as an exciting exploration of the effects of exercise on monocytes and macrophages, and their role in sepsis-induced cardiomyopathy (SICM). The reviewer's acknowledgment of the elegant and complementary tools utilized in our study is highly encouraging. We also greatly value and respect the constructive comments from this expert, and we are confident that, as outlined below, the resolution of these limitations can reach the further estimation of this reviewer and future readers. Once again, we sincerely thank the reviewer for his/her insightful comments, which have significantly contributed to enhancing the quality and translational relevance of our manuscript. We sincerely hope that the revised manuscript addresses all concerns and provides valuable insights for future readers.

Specific questions and concerns:

1. The effects on monocytes are likely acute and fast since these cells are short-lived? How can this be aligned with the observed effects on macrophages within the heart? The authors now perform a 3-month study, but in principle if exercise-induced lactate indeed mediates the effects via monocytes, one session of exercise should elicit similar effects. Such experiment with shorter time points would help to support the current working scheme and hypothesis. Another aspect that should be explored is if exercise induces shifts in tissue resident macrophages as they can also contribute to the observed effects. Importantly, the effects of exercise should also be explored before LPS stimulation.

Response: We thank the reviewer for raising these important points about the

time course of monocyte-mediated protection, the potential involvement of tissue-resident macrophages, and the effects of exercise prior to LPS stimulation. We think here the reviewer has touched on a very important point. Following the reviewer's insightful comments, we conducted additional experiments. Our new experiments demonstrate that monocytes require at least 14 days to significantly increase histone lactylation at H3K18, with levels continuing to rise until 28 days. Additionally, *in vivo* experiments demonstrate that exercised monocytes rather than resident macrophages protect cardiac function upon SICM. Indeed:

i. In response to the reviewer's comment, we investigated the duration of exercise required to induce histone lactylation in monocytes and its protective effects on cardiac function during SICM. A time-course analysis revealed that Pan-K1a and H3K181a levels remained comparable in monocytes from mice after 1 or 7 days of exercise but significantly increased after 14 days of exercise, with even higher levels seen at 28 days of exercise (Figure S10A-C). In line with these observations, 14 days of exercise mitigated body weight loss and preserved cardiac functions upon LPS challenge, while prolonged exercise for 28 days further enhanced these protective effects (Figure S10D-G). The reviewer is correct about the short lifespan of monocytes, which is approximately 1 to 3 days¹³. Our time course results hint that a single session of exercise only boosts histone lactylation in a certain portion of monocytes, and due to their short lifespan, the histone lactylation ratio declines before the next round of exercise. Thus, significant histone lactylation in monocytes requires multiple sessions of exercise, with a minimum duration of 14 days to observe robust effects. After 28 days of exercise, histone lactylation in monocytes reaches a peak and remains stable even with continued exercise. Upon LPS challenge, these monocytes with enhanced histone lactylation were recruited into cardiac tissue to protect cardiac functions. These findings, prompted by the reviewer's comments, highlight the time-dependent nature of epigenetic reprogramming in newly recruited monocytes, which is necessary to achieve robust histone lactylation and subsequent gene transcription.

ii. To address the reviewer's concern about the potential involvement of tissue resident macrophages, we performed adoptive-transfer experiments using monocytes isolated from one-month exercised donor mice, which were infused into sedentary recipients post-sepsis induction. Monocytes isolated from exercised mice exhibited increased glycolysis compared to those from sedentary mice (Figure S12L-N). Monocyte transfusion significantly reduced the drop in body temperature and body weight, as well as IL-1 β and TNF- α production caused by sepsis (Figure 9A-D). Furthermore, monocyte transfusion from exercised mice significantly improved cardiac function and reduced apoptosis in the hearts of SICM mice (Figure 9E-I). These results demonstrate that the training of circulating monocytes by exercise, rather

than resident macrophages, plays a central role in protecting against SICM.

iii. To further confirm the protective effects of histone lactylation in monocytes against SICM, we assessed the *in vivo* impact of lactylation-inhibited monocytes. Monocytes were pretreated with C646, a selective inhibitor of p300, which is the identified lactyltransferase for histone lactylation induced by exercise⁷, before adoptive transfer into SICM mice. Mice receiving p300-inhibited monocytes exhibited an abrogation of the protective effects of exercise, including loss of improvements in body temperature, weight loss, cardiac function, and myocardial apoptosis (Figure S13A-J). These findings indicate that histone lactylation in monocytes, with p300 as the writer, plays a critical role in protecting cardiac functions in SICM. This further supports the conclusion that circulating monocytes, rather than resident macrophages, mediate the protective effects of exercise against SICM.

In summary, the new data in response to the reviewer's comments have been updated in this revised manuscript. These findings provide critical insights into the time-dependent effects of exercise on monocytes, the role of circulating monocytes versus tissue-resident macrophages, and the importance of histone lactylation in mediating exercise-induced protection against SICM.

2. Related to the latter point, key controls (PBS + Exe) are missing in quite some of the assays (e.g. bulk and scRNAseq) and as such once cannot dissect which effects are truly induced by exercise perse. These data are crucial to answer the key question of how monocyte and macrophage profiles are modulated by exercise (before induction of sepsis). It looks like the sc analyses did not reveal any significant differences. Were ECAR levels also increased in exercised mice without sepsis? If that is not the case, I also don't see the point of lactate-education and monocyte transfer.

Response: This reviewer has raised an excellent and important point, and we sincerely thank him/her for this insightful comment. To this end, we have used additional approaches:

i. In this study, we begin with bulk RNA-seq analysis on PBS (Sed), PBS (Exe), LPS (Sed), and LPS (Exe) mice, and the data suggest that regular exercise mitigates the excessive immune activation caused by SICM while restoring transcriptional pathways essential for cardiac function that are disrupted by SICM (Figure 1N). These observations led us to further characterize the cardiac immune microenvironment in exercised septic hearts by isolating the CD45⁺ cell populations and conducting single-cell RNA sequencing (scRNA-seq) analysis on murine hearts. Due to bulk RNA-seq analysis, we found that exercise alone (in the absence of sepsis) caused minimal changes in transcription levels. Using conventional criteria $|\log_{2}FC| > 1$ and $\text{adj.pvalue} < 0.05$, exercise alone significantly upregulated only

two genes (Marco and Gbp11) and downregulated 12 genes (Cd300lf, Fgr, Mmp8, Ccl22, Vwa3b, Cd177, Hspa1a, Hsd11b2, Slc7a11, Fndc7, Nr4a3, Mcemp1). In contrast, under sepsis conditions, exercise resulted in the significant upregulation of 300 genes and the downregulation of 147 genes (Figure 1M). This indicates that the impact of regular exercise on the heart at the transcription level is very limited under normal conditions. Furthermore, various indicators of cardiac function and myocardial injury showed limited differences between the exercise and non-exercise groups in the absence of sepsis. Considering this, along with the high cost of the scRNA-seq analysis in 2021, we decided to exclude the PBS+Exe group from subsequent scRNA-seq analysis. However, the reviewer's concerns have made us realize that it was unreasonable to delete this control group without providing an explanation. Therefore, we have now included information about the PBS+Exe group, performed a comparative analysis in the bulk RNA-seq data (Figure 1L-N), and revised the manuscript to better clarify the effects of exercise without sepsis.

ii. In addition, to address this reviewer's sixth comment, we have performed immune infiltration analysis on bulk RNA-seq data using ImmuCellAI. The observations are consistent with the results of scRNA-seq analysis, showing no significant differences between the PBS (Sed) and PBS (Exe) groups (Figure S1). These findings further support that the absence of a PBS (Exe) control group in scRNA-seq analysis does not compromise the validity of the study.

iii. Following this reviewer's comments, we have also measured ECAR levels in monocytes from sedentary and exercised mice without sepsis. Monocytes from exercised mice exhibited significantly higher glycolysis compared to those from sedentary mice (Figure S12L-N). This indicates that exercise induces metabolic reprogramming in monocytes, even in the absence of sepsis. Consistently, both Pan-K1a and H3K181a levels were elevated in monocytes from exercised mice compared to sedentary controls treated with PBS (Figure 7A-C). Similar increases in lactylation were also observed in monocytes isolated from exercised humans (Figure 7N-P). Moreover, as described above, transfusion of monocytes from exercised donor mice into sedentary recipients post-sepsis induction significantly preserved sepsis and cardiac function (Figure 9A-I). These findings further support the protective role of exercise-trained monocytes during sepsis. The observed increase in glycolysis and histone lactylation supports the concept of lactate education. Exercise pre-adapts monocytes by enhancing their metabolic and histone lactylation, which pre-adapts them for protective roles during subsequent sepsis challenges.

3. While LPS is often used to mimic sepsis in vivo, it does not truly induce sepsis. Therefore, key findings should be validated in a proper sepsis model to support

the claims being made (e.g. clp and/or cecal slurry injection)

Response: We appreciate the reviewer's point, and in response, we utilized a more clinically relevant model, the cecal ligation and puncture (CLP) model, to induce SICM in polymicrobial sepsis¹². Consistent with the observations in the LPS-induced model, exercise significantly prevented the drop in body temperature, weight loss, and IL-1 β and TNF- α levels in CLP-induced SICM (Figure S12A-D). Additionally, exercised mice showed significantly improved cardiac function, and reduced apoptosis in the hearts of SICM mice (Figure S12E-K). These results further demonstrate that the training of circulating monocytes by exercise plays a central role in protecting against SICM.

4. Based on a drop in IL-6 and IL-10, the authors claim an attenuated pro-inflammatory response. Yet, IL-10, and to some extent also IL-6, are anti-inflammatory cytokines. As such, key inflammatory mediators like IL-1b, TNF etc should be measured to support this claim. Strikingly, IL-6 is also a myokine induced by muscle and it is therefore surprising to see it being reduced in the Exe group. Maybe timing wasn't ideal? The authors should at least discuss this aspect.

Response: We think here the reviewer touched on a very tricky point. And we thank the reviewer for this insightful comment. As a myokine, IL-6 is released during exercise in the absence of TNF- α and IL-1 β secretion¹⁴, unlike leukocyte-derived IL-6¹⁵. To provide a more comprehensive characterization of the inflammatory response, we have now measured two canonical pro-inflammatory mediators, IL-1 β and TNF α in plasma using ELISA. Consistent with previous observations, IL-1 β and TNF- α levels are significantly lower in exercised mice compared to sedentary controls following LPS challenge (Figure 1O, P). In addition, measurement of IL-1 β and TNF- α levels in various experimental setting, including Arg1 inhibition experiment (Figure S6H, I), the CLP model (Figure S12C, D), and the monocyte transfusion from exercised mice to sedentary mice (Figure 9C, D), further supports the conclusion that exercise attenuates the pro-inflammatory response in SICM. Regarding IL-6, we acknowledge its dual role as both a cytokine and a myokine. The peak level of myokine-derived IL-6 typically occurs at the end of exercise or shortly thereafter, followed by a rapid decline toward pre-exercise levels¹⁶. We believe that our sampling time point reflects the systemic septic response rather than the transient post-exercise IL-6 surge, which may explain the observed reduction in IL-6 levels in the Exe group upon LPS challenge. We appreciate this reviewer's comments, which will help us further address the potential influence of muscle-derived IL-6 during exercise. To fully exclude this possibility, we have included measurements of IL-1 β and TNF- α concentrations in Figure 1 O, P.

5. While the authors do show the importance of iNOS in mediating the observed effects (using an elegant approach with inhibitor/nanoparticle), such proof for Arg1 is missing and should be provided to support all claims being made. The overall working hypothesis is that lactate-induced monocytes from the circulation end up in the heart and become Arg1-expressing cells that do the job? Is there actual proof of this? There are very good anti-Arg1 antibodies available for flow cytometry so it would be relatively easy to see if transferred monocytes become Arg1+ cells. Actual proof should come from the use of KO models to support this hypothesis. Now Arg1 is rather a marker, and not yet a mediator.

Response: We thank the reviewer for this valuable suggestion and acknowledge that a more in-depth analysis of the cardioprotective effects of Arg1 induced by exercise and exercise-induced lactate in monocytes would be more convincing. To this end, we have used additional approaches.

i. To further investigate the role of Arg1 in monocytes during SICM, we employed the previously described nanoparticle to specifically inhibit Arg1 expression in Ly6C⁺ monocytes (Figure 4C, Figure S6E-G). As expected, Arg1 inhibition abolished the beneficial effects of exercise on reducing the decline in body temperature, weight loss, IL-1 β and TNF- α levels, and sepsis-induced myocardial injury (Figure 4T-X, Figure S6H-L). These findings demonstrate that Arg1 in monocyte-derived macrophages is critical for protecting cardiac function during SICM.

ii. Furthermore, to demonstrate that Arg1 expression is induced by exercise via lactate accumulation, we performed flow cytometric analyses using the anti-Arg1 antibody recommended by the reviewer to quantify Arg1 expression in cardiac macrophages. In exercised mice, there was a significant increase in Arg1⁺ cardiac macrophages compared to sedentary mice, however, in mice with a myeloid-specific knockout of LDHA, the number of Arg1⁺ cardiac macrophages was markedly reduced compared to exercised wild-type controls (Figure S9D). Consistently, adoptive transfer of lactate-educated monocytes into naïve recipients resulted in a significant increase in Arg1⁺ macrophages in the heart, whereas Arg1 inhibition led to impaired cardiac function in SICM (Figure 4T-X, Figure S6H-L). Together, these data confirm that Arg1 expression in monocyte-derived macrophages is indispensable for the cardioprotective effects of exercise in SICM and further validate the essential role of LDHA in inducing Arg1 expression.

iii. To investigate how exercise-induced histone H3K18la (Figure 7A-C) regulates *Arg1* expression, we performed genome-wide Cleavage Under Targets and Tagmentation (CUT&Tag) analysis using an anti-H3K18la antibody, followed by high-throughput DNA sequencing. The data revealed significant enrichment of H3K18la in the promoter regions of *Arg1* (Figure

8P). Additionally, CUT&Tag-qPCR analysis indicated increased H3K18la enrichment in the promoter regions of Arg1 following LPS stimulation compared to PBS injection, and to a higher extent in the context of regular exercise (Figure 8Q).

In summary, these results demonstrate that Arg1 expression in monocytes plays a critical role in the cardioprotective effects of exercise during SICM. Furthermore, Arg1 expression is promoted through exercise-induced H3K18 histone lactylation in monocytes.

6. Bulk sequencing appears a bit redundant since the authors also performed scRNAseq. The rationale of using both should be better explained, and also similarities and differences in the results should be highlighted. With regards to differences being observed in the indicated genes in bulk, analyses, it would be important to use the sc data to dissect which cell types are responsible for the observed effects in bulk analyses

Response: We thank the reviewers for their insightful comments and suggestions. We acknowledge that our initial analysis did not adequately highlight the complementary value of bulk RNA-seq in comparison to scRNA-seq. To address this concern, we have revised our manuscript to better explain the rationale for using both methodologies and to highlight the similarities and differences between the results obtained from bulk RNA-seq and scRNA-seq analyses. As described in our response to the reviewers' second comment, we have supplemented the analysis by including the control group (PBS+Exe). In the updated version of Figure 1, we have replaced Figure 1L-N to provide a more comprehensive analysis. Specifically, PCA analysis and differential gene count statistics demonstrate that under PBS conditions, regular exercise for 3 months barely induces changes in transcription levels, whereas the effects of exercise are predominantly observed under sepsis conditions (Figure 1L, M). This observation explains why the PBS+Exe control group was not included in the scRNA-seq analysis. Furthermore, Figure 1N illustrates that bulk RNA-seq results reveal the primary effect of exercise under sepsis conditions is to inhibit excessive immune responses and restore myocardial function. These findings provided the rationale for selecting CD45⁺ cells for subsequent scRNA-seq analysis. In contrast, scRNA-seq offers higher resolution and reveals additional insights. Specifically, scRNA-seq identifies monocyte-derived macrophages as the key cell population responsible for restoring cardiac immune homeostasis in SICM under exercise conditions. This highlights the complementary nature of the two methodologies, with bulk RNA-seq providing a broader overview of transcriptional changes and scRNA-seq enabling cell-type-specific dissection of the observed effects. To address the reviewer's suggestion regarding the dissection of cell types responsible for the observed effects in bulk analyses, we performed immune infiltration analysis on bulk RNA-seq data using ImmuCellAI. As shown in Figure S1, under SICM conditions, exercise increases

infiltration of macrophages, monocytes, and neutrophils, which aligns with the observations in scRNA-seq analysis. In addition, under non-SICM conditions, exercise does not cause significant immune infiltration changes, which provides a basis for the conclusion that the absence of a PBS+Exe control group in scRNA-seq analysis does not affect the validity of the study and further addresses the reviewers' second comment. We have revised the results section to incorporate these findings and expanded the discussion to better describe the similarities and differences between bulk RNA-seq and scRNA-seq methodologies.

7. The authors now did a transfer of lactate-educated monocytes. Transfer monocytes from trained mice to naive ones before the induction of sepsis should be performed to demonstrate that it are indeed the training of monocytes mediating the effects. This is important as also resident macrophages will be influenced by the exercise and likely also contribute to the observed phenotype.

Response: We thank the reviewer for this insightful suggestion. In response, we conducted adoptive-transfer experiments in which monocytes isolated from exercised donor mice were infused into sedentary recipients following sepsis induction. Monocyte transfusion significantly alleviated the drop in body temperature and body weight, as well as reduced IL-1 β and TNF- α production associated with sepsis (Figure 9A-D). Furthermore, monocyte transfusion from exercised mice significantly improved cardiac function and reduced apoptosis in the hearts of SICM mice (Figure 9E-I). In summary, these results demonstrate that the training of circulating monocytes by exercise, rather than resident macrophages, plays a central role in protecting against SICM.

8. The authors mention that resident macrophage markers like Apoe, C1qa, Mrc1 were upregulated during differentiation from Ly6c Mo to MCHII hi macrophages. I would say monocytes do not differentiate into tissue-resident macrophages so (i) the markers are not good markers, and/or (ii) the pseudotime analysis makes a trajectory that is not actually there.

Response: We sincerely appreciate the reviewer's insightful comments and the opportunity to address this critical point. As the reviewer has highlighted, the differentiation of monocytes into tissue-resident macrophages remains a complex and widely debated topic. On one hand, the conventional classification of cardiac macrophages is into tissue-resident macrophages and monocyte-derived macrophages. Meanwhile, emerging studies suggest that cardiac macrophages can be categorized into three groups based on their renewal patterns: self-renewing resident macrophages with negligible blood monocyte input (Lyve1⁺ M \emptyset), monocytes partially replacing macrophages (MHC II^{high} M \emptyset), and fully replaced macrophages (Ccr2⁺ M \emptyset)^{17,18}. Under this updated framework, the traditional tissue-resident macrophages correspond to Lyve1⁺ M \emptyset , which supports the reviewer's statement that monocytes do not differentiate into

tissue-resident macrophages. In agreement with this reviewer's comment, we excluded Lyve1⁺ Mø classical resident macrophages from the proposed pseudotime trajectory, ensuring that resident macrophages were not included in this analysis. We greatly appreciate the reviewer's feedback, which has helped us recognize that our description of 'tissue residence' in the proposed pseudotime trajectory may have been misleading. We also acknowledge that Apoe, C1qa, and Mrc1 are not classic markers of tissue-resident macrophages. To address these concerns, we have revised the manuscript and rephrased the referenced sentence as follows 'Analysis of the representative genes and functional enrichment for each pattern revealed that genes related to interleukin-1 production, regulation of T cell activation, and leukocyte migration were gradually upregulated during the differentiation from Ly6c2^{high} Mo to MHC II^{high} Mø.' on Page 11 Line 9. Furthermore, the differentiation trajectories of Ly6c Mo to MHC II high macrophages have been reported in previous studies^{17,18}. We regret that our phrasing may have inadvertently led the reviewer to assume that MHC II high Mø were tissue-resident macrophages. We sincerely apologize for this misunderstanding and believe that the revised manuscript and this clarification will effectively address the reviewer's concerns.

9. Ldha show decreased Klf4 but also other effects. To what extent does the lactylation play a role? Further (albeit also indirect) proof could be provided by using inhibitors that block histone lactylation.

Response: We thank the reviewer for this insightful comment. To further confirm the protective effects of histone lactylation in monocytes against SICM, we assessed the *in vivo* impact of lactylation-inhibited monocytes. Monocytes were pretreated with C646, a selective inhibitor of p300, which is the identified lactyltransferase for histone lactylation induced by exercise⁷, before adoptive transfer into SICM mice. Mice receiving p300-inhibited monocytes exhibited an abrogation of the protective effects of exercise, including loss of improvements in body temperature, weight loss, cardiac function, and myocardial apoptosis (Figure S13A-J). These findings indicate that histone lactylation in monocytes, with p300 as the writer, plays a critical role in protecting cardiac functions in SICM.

10. Overall, the human relevance of the current findings are very limited and key aspects should be validated in the human setting. For example; is lactylation also increased in human monocytes by exercise?

Response: We thank the reviewer for raising this important point about clinical translation. To address this, we recruited volunteers with sedentary and active lifestyles (Figure 7J, Table S3). Given that lactate levels influence histone lactylation, as reported in previous studies^{3,4}, we first exposed circulating monocytes from mice to varying concentrations of lactate to induce histone lactylation and identified 20 mM lactate as the optimal concentration (Figure

S9E, F). Building on this, we treated monocytes from sedentary volunteers with 20mM exogenous lactate and observed increased levels of Pan-K1a and H3K181a (Figure 7K-M). Furthermore, monocytes from active volunteers exhibited significantly higher levels of both Pan-K1a and H3K181a compared to those from sedentary individuals (Figure 7N-P). In summary, these findings demonstrate that exercise-induced H3K18 histone lactylation is conserved in human monocytes, underscoring the translational relevance of our study.

References:

- 1 Zhao, L., Qi, H., Lv, H. *et al.* Lactylation in health and disease: physiological or pathological? *Theranostics* **15**, 1787-1821 (2025).
- 2 Zhang, D., Tang, Z., Huang, H. *et al.* Metabolic regulation of gene expression by histone lactylation. *Nature* **574**, 575-580 (2019).
- 3 Niu, Z., Chen, C., Wang, S. *et al.* HBO1 catalyzes lysine lactylation and mediates histone H3K91a to regulate gene transcription. *Nat Commun* **15**, 3561 (2024).
- 4 van Amerongen, M. J., Harmsen, M. C., van Rooijen, N., Petersen, A. H. & van Luyn, M. J. Macrophage depletion impairs wound healing and increases left ventricular remodeling after myocardial injury in mice. *Am J Pathol* **170**, 818-829 (2007).
- 5 Li, F., Si, W., Xia, L. *et al.* Positive feedback regulation between glycolysis and histone lactylation drives oncogenesis in pancreatic ductal adenocarcinoma. *Mol Cancer* **23**, 90 (2024).
- 6 Shang, M., Cappellesso, F., Amorim, R. *et al.* Macrophage-derived glutamine boosts satellite cells and muscle regeneration. *Nature* **587**, 626-631 (2020).
- 7 Zhang, N., Zhang, Y., Xu, J. *et al.* alpha-myosin heavy chain lactylation maintains sarcomeric structure and function and alleviates the development of heart failure. *Cell Res* **33**, 679-698 (2023).
- 8 Moreno-Yruela, C., Zhang, D., Wei, W. *et al.* Class I histone deacetylases (HDAC1-3) are histone lysine delactylases. *Sci Adv* **8**, eabi6696 (2022).
- 9 Bahador, M. & Cross, A. S. From therapy to experimental model: a hundred years of endotoxin administration to human subjects. *J Endotoxin Res* **13**, 251-279 (2007).
- 10 Alvarez, S., Vico, T. & Vanasco, V. Cardiac dysfunction, mitochondrial architecture, energy production, and inflammatory pathways: Interrelated aspects in endotoxemia and sepsis. *Int J Biochem Cell Biol* **81**, 307-314 (2016).
- 11 Jia, L., Wang, Y., Wang, Y. *et al.* Heme Oxygenase-1 in Macrophages Drives Septic Cardiac Dysfunction via Suppressing Lysosomal Degradation of Inducible Nitric Oxide Synthase. *Circ Res* **122**, 1532-1544 (2018).
- 12 Rittirsch, D., Huber-Lang, M. S., Flierl, M. A. & Ward, P. A. Immunodesign of experimental sepsis by cecal ligation and puncture. *Nat Protoc* **4**, 31-36 (2009).
- 13 Patel, A. A., Zhang, Y., Fullerton, J. N. *et al.* The fate and lifespan of human monocyte subsets in steady state and systemic inflammation. *J Exp Med* **214**, 1913-1923 (2017).
- 14 Petersen, A. M. & Pedersen, B. K. The anti-inflammatory effect of exercise. *J Appl Physiol (1985)* **98**, 1154-1162 (2005).
- 15 Kistner, T. M., Pedersen, B. K. & Lieberman, D. E. Interleukin 6 as an energy allocator in muscle tissue. *Nat Metab* **4**, 170-179 (2022).
- 16 Pedersen, B. K. Muscular interleukin-6 and its role as an energy sensor. *Med Sci Sports Exerc* **44**, 392-396 (2012).
- 17 Dick, S. A., Macklin, J. A., Nejat, S. *et al.* Self-renewing resident cardiac macrophages limit adverse remodeling following myocardial infarction. *Nat Immunol* **20**, 29-39 (2019).
- 18 Dick, S. A., Wong, A., Hamidzada, H. *et al.* Three tissue resident macrophage subsets coexist across organs with conserved origins and life cycles. *Sci Immunol* **7**, eabf7777 (2022).

Response to Reviewer 1:

In the revised manuscript, the authors have made commendable efforts to improve their data, (especially for in-depth analysis of the characterization of exercise-induced histone lactylation). By optimizing the experimental design and supplementing key data, the authors have effectively clarified the detailed mechanisms underlying the core findings of the study. In particular, the logical chain of the core regulatory axis-"Exercise-induced lactate triggered H3K18la, effectively pre-trains monocyte derived macrophages to preadapt to sepsis before cardiac recruitment in SICM"-has become more comprehensive. Most of my previous concerns have been addressed, but a few issues remain that I believe still need attention:

Response: We sincerely thank the reviewer for defines our revision have effectively clarified the detailed mechanisms of the study. We are particularly delighted that the reviewer considers the core regulatory axis of our work to be more comprehensive. We deeply appreciate the positive feedback and recognition of our efforts to improve the data from this expert. We also highly value the reviewer's constructive comments, which have been instrumental in refining our work. As detailed below, we have thoroughly addressed the remaining concerns, and we hope that these resolutions will further enhance the manuscript and meet the expectations of both the reviewer and future readers.

1. Although this study has included sepsis patients and analyzed the metabolic characteristics of their monocytes, it has not detected the expression of Pan-Kla and H3K18la in monocytes from sepsis patients (especially those with concurrent SICM). Nor has it established associations between these lactylation indicators and patients' cardiac function prognosis (e.g., ejection fraction [EF] value, mortality rate). This leads to a lack of key correlative evidence for the conclusion that "lactylation regulation improves SICM" in the pathological context of human sepsis, thereby weakening the persuasiveness of its clinical translation. The author needs to provide some explanations in discussion.

Response: We sincerely thank the reviewer for his/her thoughtful comments and for highlighting this important aspect of our study. We fully acknowledge the limitations regarding the lack of direct detection of Pan-Kla and H3K18la expression in monocytes from sepsis patients, particularly those with concurrent SICM, as well as the absence of established correlations between these lactylation indicators and cardiac function outcomes.

In response to this valuable feedback, we have revised the Discussion section to explicitly address these limitations. Specifically, we have noted that while our study demonstrated significantly higher levels of Pan-Kla and H3K18la in monocytes from active volunteers compared to sedentary individuals, consistent with findings from the mouse model, technical constraints prevented

us from fully assessing the expression of the histone lactylation levels in monocytes from SICM patients. This limitation restricted our ability to establish direct correlations between lactylation levels and cardiac function prognosis in SICM patients, which would have further strengthened the translational relevance of our findings.

In the revised Discussion, we emphasized that future studies are essential to overcome these challenges. Specifically, we propose that future research should focus on directly investigating lactylation markers in sepsis patients with SICM and correlating their expression levels with clinical outcomes such as cardiac function and mortality rates.

We sincerely appreciate the reviewer's insightful comments, which have greatly helped us improve the manuscript by refining the Discussion section to acknowledge and address this limitation. We thank the reviewer once again for his/her valuable input, which has helped us improve the manuscript.

2. There are many elementary mistakes in this manuscript, which might leave a bad impression on this work. The authors should carefully check this manuscript and revised it. For example:

a) There is inconsistency in the expression of the core term "monocyte derived macrophages" throughout the manuscript—hyphens are omitted in some paragraphs (where it is written as "monocyte-derived macrophages"). It is recommended to standardize the term uniformly to comply with academic writing conventions and avoid ambiguity.

b) There is a term confusion between "cardiovascular diseases (CVD)" in the abstract and "cardiovascular diseases (CAD)" in the introduction. CVD (Cardiovascular Diseases) is a general term for cardiovascular disorders (and SICM, i.e., sepsis-induced cardiomyopathy, falls within this category), while CAD specifically refers to Coronary Artery Disease. These two concepts are distinct, which may easily cause readers to misunderstand the scope of the study. It is necessary to uniformly revise the term to "cardiovascular diseases (CVD)".

Response: We sincerely thank the reviewer for his/her careful and insightful comments, which have greatly contributed to improving the accuracy and presentation of our manuscript. In response to these valuable suggestions, we conducted a thorough line-by-line review of the entire manuscript and included the following revisions:

a) We standardized the term "monocyte-derived macrophages" throughout the manuscript to ensure uniformity, consistency, and adherence to academic writing conventions, thereby eliminating any ambiguity in its usage.

b) We corrected all instances of "CAD" to "CVD" in the Introduction and throughout the manuscript. This ensures that the term "cardiovascular diseases (CVD)" is consistently used, avoiding confusion between distinct concepts such

as Coronary Artery Disease (CAD).

Furthermore, we carefully addressed all other editorial inconsistencies and elementary mistakes identified during this review process. Following this reviewer's valuable feedback, we believe these revisions have resolved the issues raised, improved clarity, and enhanced the overall quality of the manuscript.

Response to Reviewer 2:

The authors addressed to most important concerns and I now consider this article suitable for publication.

Response: We sincerely thank the reviewer for his/her thoughtful evaluation of our manuscript and for providing constructive suggestions during the previous review process. These comments have been invaluable in helping us improve the quality of our study. We are truly grateful for his/her recognition of our efforts and for considering this article suitable for publication.

Response to Reviewer 1:

The authors have added adequate descriptions to answer my queries. I think this paper can now go to the next step for publication.

Response: We sincerely thank the reviewer for his/her thoughtful evaluation of our manuscript and for providing constructive feedback throughout the review process. His/her insightful comments have been instrumental in improving the quality of our study. We deeply appreciate his/her recognition of our efforts and are truly grateful for considering this article suitable for publication.